# Unveiling the anti-obesity potential of Kemuning (*Murraya paniculata*): A network pharmacology approach

Rizka Fatriani[1], Firda Agustin Kartika Pratiwi[2], Annisa Annisa[2], Dewi Anggraini Septaningsih[3], Sandra Arifin Aziz[4], Isnatin Miladiyah[5], Siska Andrina Kusumastuti[6], Mochammad Arfin Fardiansyah Nasution[7], Donny Ramadhan[6], Wisnu Ananta Kusuma[1,2]*

1 Tropical Biopharmaca Research Center, IPB University, Bogor, Indonesia, 2 Department of Computer Science, Faculty of Mathematics and Natural Sciences, IPB University, Bogor, Indonesia, 3 Department of Chemistry, Faculty of Military Mathematics and Natural Sciences, Republic of Indonesia Defense University, Bogor, Indonesia, 4 Department of Agronomy and Horticulture, Faculty of Agriculture, IPB University, Bogor, Indonesia, 5 Faculty of Medicine, Islamic University of Indonesia, Indonesia, 6 Research Center for Pharmaceutical Ingredients and Traditional Medicine, National Research and Innovation Agency (BRIN), Bogor, Indonesia, 7 Department of Chemistry, Faculty of Mathematics and Natural Sciences, Universitas Indonesia, Depok, Indonesia

* ananta@apps.ipb.ac.id

**Data Availability Statement:** All relevant data are within the paper and its supporting information files.

## Abstract

Obesity has become a global issue that affects the emergence of various chronic diseases such as diabetes mellitus, dysplasia, heart disorders, and cancer. In this study, an integration method was developed between the metabolite profile of the active compound of *Murraya paniculata* and the exploration of the targeting mechanism of adipose tissue using network pharmacology, molecular docking, molecular dynamics simulation, and in vitro tests. Network pharmacology results obtained with the skyline query technique using a block-nested loop (BNL) showed that histone acetyltransferase p300 (EP300), peroxisome proliferator-activated receptor gamma (PPARG), and peroxisome proliferator-activated receptor gamma coactivator 1-alpha (PPARGC1A) are potential targets for treating obesity. Enrichment analysis of these three proteins revealed their association with obesity, thermogenesis, energy metabolism, adipocytokines, fat cell differentiation, and glucose homeostasis. Metabolite profiling of *M. paniculata* leaves revealed sixteen active compounds, ten of which were selected for molecular docking based on drug-likeness and ADME results. Molecular docking results between PPARG and EP300 with the ten active compounds showed a binding affinity value of $\leq$ -5.0 kcal/mol in all dockings, indicating strong binding. The stability of the protein-ligand complex resulting from docking was examined using molecular dynamics simulations, and we observed the best average root mean square deviation (RMSD) of 0.99 Å for PPARG with trans-3-indoleacrylic acid, which was lower than with the native ligand BRL (2.02 Å). Furthermore, the RMSD was 2.70 Å for EP300 and the native ligand 99E, and the lowest RMSD with the ligand (1R,9S)-5-[(E)-2-(4-Chlorophenyl) vinyl]-11-(5-pyrimidinylcarbonyl)-7,11-diazatricyclo[7.3.1.02,7]trideca-2,4-dien-6-one was 3.33 Å. The in vitro tests to validate the potential of *M. paniculata* in treating obesity showed that there was a significant decrease in PPARG and EP300 gene expressions in 3T3-L1

**Funding:** This research was supported by the Ministry of Research, Technology and Higher Education, Indonesia, under the Competitive Research Grant from the Directorate of Higher Education, Indonesia, in 2023. No. 102/E5/PG.02.00.PL/2023. URL: https://dikti.kemdikbud.go.id/. The author who received the grant is WAK. The funders had no role in the study design, data collection and analysis, decision to publish, or manuscript preparation.

**Competing interests:** The authors have declared that no competing interests exist.

mature adipocytes treated with *M. paniculata* ethanolic extract starting at concentrations 62.5 µg/ml and 15.625 µg/ml, respectively. These results indicate that *M. paniculata* can potentially treat obesity by disrupting adipocyte maturation and influencing intracellular lipid metabolism.

## Introduction

In 2016, 1.9 billion adults over the age of 18 (39%) were overweight worldwide, and 650 million (13%) were in the obesity category; this figure tripled compared to 1975. The incidence of overweight and obesity in children and adolescents (aged 5–19 years) is 350 million (18%), and this percentage has increased dramatically from 4% (1975) to 18% (2016) [1]. In 2030, this figure is expected to be 2.16 billion for overweight cases and 1.12 billion for obesity cases [2]. Obesity contributes significantly to the incidence of various diet-related chronic diseases, including diabetes mellitus, dyslipidemia, heart and blood vessel disorders, organ fat, and several other disorders, including cancer. Overweight and obesity occur because of excessive fat accumulation in adipose tissue due to an energy imbalance, in which the energy consumed is higher than that expended [3]. The global pandemic of overweight and obesity and its relationship with energy imbalance requires a deeper understanding of the role of adipose tissue in regulating energy metabolism.

In recent years, the adipose tissue has become a central focus in studying energy-related metabolic disorders because it plays a significant role in the interactions between nutrition, health [4], metabolism, inflammation [5], and energy balance [4, 5]. The adipose tissue is also important for regulating glucose levels, lipid homeostasis, and body weight [6, 7]. The two large adipose tissues are white adipose tissue (WAT) and brown adipose tissue (BAT) [6, 8]. WAT stores excess energy as fat, whereas BAT removes excess fat in the form of energy [6]. Obesity can cause WAT dysfunction in calorie storage due to excess nutrition. Adipose tissue expansion is caused by an increase in the number of adipocytes (hyperplasia) and their size (hypertrophy) [9]. Massive expansion of adipose tissue in obesity impacts on blood vessel dysfunction and cardiovascular disease [10]. In addition, excess nutrition in obesity will induce an increased accumulation of ectopic lipids called lipotoxicity [9].

Various approaches, including the use of herbal plants, have been utilized to treat obesity. Kemuning (*Murraya paniculata*) is an herbal plant native to the Andaman Islands, Assam, Bangladesh, the Bismarck Archipelago, Borneo, Cambodia, South-Central China, Southeast China, Christmas Island, Eastern Himalayas, Hainan, Java, Laos, Lesser Sunda Is., Malaya, Maluku, Myanmar, Nepal, New Guinea, New South Wales, Northern Territory, Philippines, Queensland, Solomon Island, Sri Lanka, Sulawesi, Sumatra, Taiwan, Thailand, Vanuatu, Vietnam, Western Himalayas, and Western Australia [11]. This plant has various pharmacological activities, including anti-obesity [12–14], analgesic [15–17], anti-inflammatory [15, 17, 18], antioxidant [19–21], anti-diarrheal [18], anti-bacterial [22–24], and anti-hyperglycemic effects [25]. *M. paniculata* is reported to contain flavonoids [26–29], coumarins [30–33], alkaloids, saponins, and tannins [34].

The anti-obesity effect of *M. paniculata* has been reported by Suwandi et al., who administered an ethanol extract of *M. paniculata* leaves at doses of 200, 400, and 800 mg/kg to mice that had been subcutaneously induced with monosodium L-glutamate (2 g/kg) for five days and high-carbohydrate diets, and showed inhibited body weight gain. An effective dose of 200 mg/kg inhibited body weight gain by 109.28% [12]. Other studies on the anti-obesity activity

of *M. paniculata*, based on its ability to inhibit pancreatic lipase activity, showed that 100 ppm of *M. paniculata* water extract had an inhibitory effect of 25.66% [13].

Although *M. paniculata* has been widely studied as an anti-obesity agent, no comprehensive studies have explored the mechanism of *M. paniculata* as an anti-obesity agent using a network pharmacological and in vitro assay approach. This study aimed to answer these questions by investigating the mechanism of active compounds found in *M. paniculata* using a metabolomic approach and by screening target proteins associated with obesity using a network pharmacology approach. The results were validated using molecular docking, molecular dynamics simulations, and in vitro tests. In vitro tests used 3T3-L1 pre-adipocytes, which are model cells from white adipose tissue, to observe the effect of *M. paniculata* on adipose tissue development.

This network pharmacology approach has been widely used to investigate the potential of bioactive compounds from natural products and determine their therapeutic capability in certain diseases [35, 36]. In addition to using the network pharmacology approach, one of the contributions of this study is the application of skyline queries, a technique used in constrained optimization problems, to the domain of protein-protein interaction networks, especially in protein target selection. Using this technique, we identified the significant protein targets that were used to screen for potential compounds in *M. paniculata*.

## Material and methods

### Network pharmacology

**Screening of protein targets.** The protein targets were screened based on the disease and active compounds of *M. paniculata*. Protein targets associated with obesity were obtained from the OMIM database (https://www.omim.org/) [37]. Data were collected via web scrapping using Python 3.9 programming language [38]. The input keywords were "+Brown+-Adipose+Tissue".

The protein targets related to the active compounds were collected from the IJAH Analytics database (http://ijah.apps.cs.ipb.ac.id/) [39] using the keyword "*Murraya paniculata*" as the search input. This database contains plant data, most of which are from Indonesia, and compounds, target proteins, and their interactions were obtained from PubChem (https://pubchem.ncbi.nlm.nih.gov/) and UniProt (https://www.uniprot.org/). In addition, protein targets were filtered based on active compounds obtained from the STITCH database (http://stitch.embl.de/) [40]. The input entered was in the form of SMILES data from the active compounds and was limited to the "*Homo sapiens*" category.

**Construction of protein-protein interaction (PPI) network.** Targets obtained from the OMIM and STITCH databases were combined, and protein interactions were constructed using STRING (https://string-db.org/) [41]. The parameters used to search for the PPI data were the network-type full STRING network and a medium confidence score of 0.400.

Protein interactions obtained from the STRING database were analyzed using Cytoscape [42]. Subgraphs that do not interact with the main graph were omitted, and the data were transformed into a skyline query domain. Data transformation into a skyline query domain was carried out using centrality measures, which is one way to measure the influential nodes. The centrality measures used included degree, betweenness, closeness, bridging, radiality, and eigenvector centralities. Data transformation was carried out using the CentiScaPe2.2 tool in the Cytoscape application.

**Selecting target protein using skyline query technique.** A skyline query is a widely recognized technique for selecting a limited number of dominant objects. Based on the skyline concept, a significant protein in an interaction network is considered a non-dominant protein.

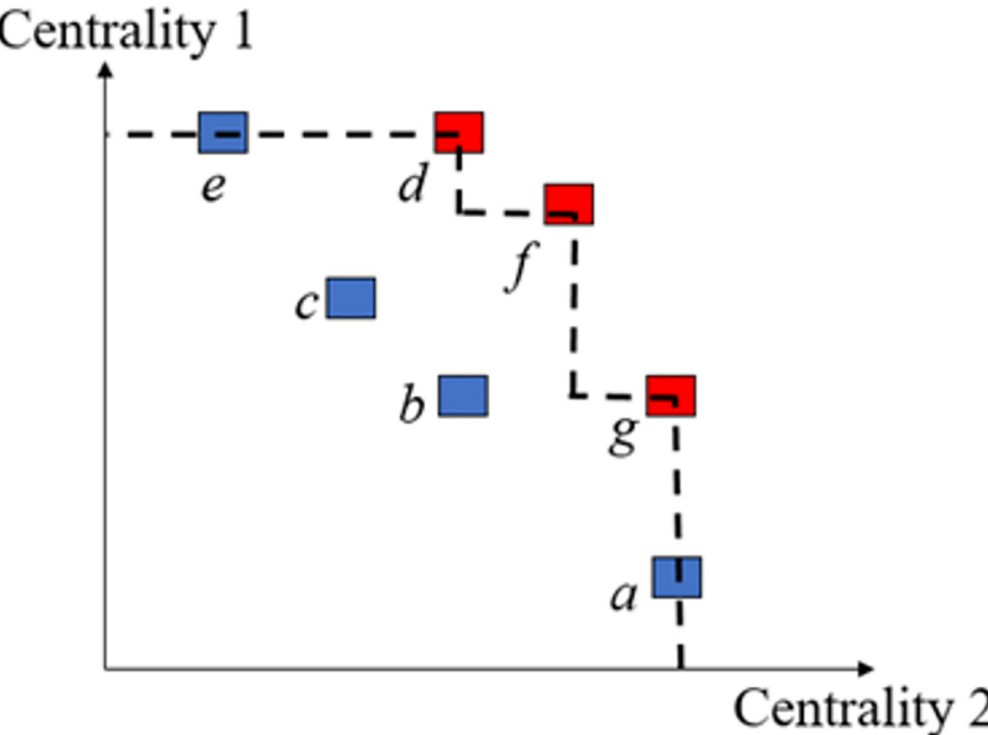

**Fig 1. Example of skyline query for proteins.**

A protein, p, is considered to dominate another protein, q, if, in each of the degree centrality values, p is either equal to or better than q and in at least one degree centrality, p is superior to q. A skyline query fetches a collection of proteins that are not dominated by another.

For example, proteins a, b, c, d, e, f, and g were mapped to two centrality values, where for both values, the better one was the larger value. In **Fig 1**, it can be observed that based on these two centrality values, protein d dominates proteins e, c, and b, protein f dominates proteins c and b, and protein g dominates proteins b and a. Proteins d, f, and g do not mutually dominate each other, and none of the proteins can dominate all three. Based on these two centrality values, significant proteins were identified as d, f, and g.

The block-nested loop (BNL) algorithm is a common approach for implementing skyline queries. This involves the use of nested loops to compare and identify non-dominated points. The "Dominates" function in the BNL is used to determine whether one point dominates another. The skyline query function then iterates through each point in the dataset and compares it with all the other points using nested loops. If a point is not dominated by any other point, it is added to the skyline. **Fig 2** shows the pseudocode of the BNL algorithm for obtaining the skyline points.

## Enrichment analysis

Enrichment analysis was performed by analyzing gene ontology (GO) and pathways. GO provides information related to gene function and products. GO is divided into three ontologies: molecular functions, biological processes, and cell components. Molecular functions are activities that occur at the molecular level and are regulated by gene products. Biological processes occur via various molecular mechanisms. Cell components are the relative locations in the cell

```
function skyline query(data points):
    skyline_set = empty set

    // Outer loop
    for each point_p in data_points:
        is dominated = false

        // Inner loop
        for each point q in data_points:
            if point_p ≠ point_q:
                // Check if point p dominates point q
                if dominates(point p, point q):
                    is_dominated = true
                    break  // Exit the inner loop if point p dominates point q

        // If point_p is not dominated, add it to the skyline set
        if not is dominated:
            add point p to skyline set

    return skyline set

function dominates(point_p, point_q):
    // Implement the dominance check based on desired criteria
    // For example, compare each dimension of the points
    for each dimension in dimensions:
        if point p[dimension] < point q[dimension]:
            return false
    return true
```

**Fig 2. Pseudocode of block-nested loop (BNL).**

structure where gene products carry out their activities [43]. The Reactome [44], Bioplanet [45], and Kyoto Encyclopedia of Genes and Genomes [46] pathways were analyzed. GO and pathway analyses were performed using the web-based application Enrichr (https://maayanlab.cloud/Enrichr/) [47]. Enrichment results were visualized using SRPlot [48] (https://www.bioinformatics.com.cn/en).

## Metabolite profiling

**M. paniculata leaves extract.** *M. paniculata* leaves were obtained from Cikabayan Experimental Garden, Tropical Biopharmaca Research Center, IPB University. 1 g of Simplicia from *M. paniculata* leaves was weighed and extracted using 10 ml of water, 50% ethanol, and 100% ethanol using the sonication method for 30 min, left for 30 min, and sonicated again for 30 min. The mixture was sonicated for 1 h and filtered. The resulting filtrate was filtered with a 0.22 μm nylon membrane.

**LC-MS analysis of *M. paniculate*.** The metabolite profile of *M. paniculata* leaves was determined using an ultra-high-performance liquid chromatography-tandem Q Extractive Plus Orbitrap HRMS (Thermo Scientific, Munich, Germany). The Accucore C18 column (100 mm × 2.1 μm, particle size, 1.5 μm, Thermo Scientific™, Waltham, MA, USA) was set to 30˚C. The mobile phases were 0.1% formic acid in water (A) and 0.1% formic acid in acetonitrile (B) at a flow rate of 0.2 mL/min. The gradient system used was as follows: 0–1 min (5% B), 1–15 min (5–45% B), 15–25 min (45–95% B), 25–28 min, (95%B), 28–33 min (5%B). The MS system settings were $m/z$ = 100−1500 (mass range), resolution = 70,000, and AGC target = $1 \times \times 10^5$. The MS2 system settings were as follows: resolution = 17,500, AGC target, $1 \times \times 10^5$; and (N) CE, 18, 35, and 53. The analysis was conducted in positive ionization mode, with a sheath gas flow rate of 35, auxiliary gas flow rate of 15, sweep gas flow rate of 0, spray voltage of 3.8 KV, capillary temperature of 320˚C, S-lens RF level of 50, and auxiliary gas heater temperature of 38˚C. The sample (2 μL) was injected into the LC HRMS.

## Drug-likeness and ADME prediction

SwissADME (http://www.swissadme.ch/) [49] was used to obtain Lipinski's rule of five, GI absorption, and ligand bioavailability scores. Ligands were obtained from the metabolite profiles of the *M. paniculata* leaves. Compounds that met the requirements of Lipinksi's rule of five [50] and had high GI absorption and bioavailability of > 0.3 were chosen for molecular docking [51].

## Molecular docking

Crystal structures of peroxisome proliferator-activated receptor gamma (PPARG) (PDB ID: 7AWC) [52] and histone acetyltransferase p300 (EP300) [53] (PDB ID: 5NU5) were obtained from the RCSB Protein Data Bank (https://www.rcsb.org/) [54]. The 3D structures of the ligands were retrieved from PubChem (https://pubchem.ncbi.nlm.nih.gov/) [55]. The 3D structure of the ligands was then optimized using Avogadro2 software [56]. Proteins and ligands were prepared using AutoDockTools [57] by deleting all water molecules and adding hydrogen bonds. The location of the binding site was determined based on the positions of native ligands in the crystal structure. The grid box was adjusted to 20 × 20 × 20 (x-, y-, and z-axes). Molecular docking was performed using AutoDock Vina [58, 59] in AutoDockTools. The docking results were visualized using PyMOL (https://pymol.org/2/) [60] for 3D visualization and BIOVIA Discovery Studio [61] for 2D interactions.

## Molecular dynamics simulations

The stability of the protein-ligand complexes formed in earlier docking studies was further examined using molecular dynamics (MD) simulations. Molecular dynamics simulations were carried out using a Dell Precision 5820 Tower X-Series with an Ubuntu 20.04 (Focal Fossa) 64-bit operating system, Intel® Core™ i9-10900X CPU @ 4.60 GHz × 20 processors, 62.5 GiB memory, and an NVIDIA® Quadro RTX™ 8000 graphics card. The GROMACS 2023.02 software [62, 63] was used for this purpose. Before initiating the MD simulation, both PPARG and EP300 enzyme structures had their N- and C-termini manually capped with ACE and NME, respectively, utilizing Chimera v1.15 software [64]. Additionally, the topologies of the ligands from previous docking results and the standard ligands for each enzyme (BRL and 99E for PPARG and EP300, respectively) were generated using Gaussian03 software [65]. The AMBER ff98SB-ILDN force field [66] was employed for the system parameterization. This was followed by box preparation, adding ligands to the system, solvent addition using TIP3P water, and neutralizing the system by adding $Na^+$ and $Cl^-$. Subsequently, two iterations of energy

minimization were conducted, followed by an equilibration run comprising 100 ps of the NVT simulation and 100 ps of the NPT simulation at 300 K. Finally, a production run of 100 ns at 300 K was performed with a time step of 2 fs. For each ligand-protein system, root mean square deviation (RMSD) and root mean square fluctuation (RMSF) analyses were conducted, focusing on the top three ligands in each system and their corresponding standard ligands.

## In vitro testing

**Cell lines and culture.** 3T3-L1 preadipocyte cells were obtained from Elabscience (Texas, USA). Cells were cultured in Dulbecco's modified Eagle's medium (DMEM, Gibco, New York, USA) containing 1% penicillin/streptomycin (PS, 100 units of penicillin/mL and 100 pg streptomycin/mL; Gibco) and supplemented with 10% newborn calf serum (NBCS, Gibco). The cells were maintained at 37°C in a humidified atmosphere containing 5% $CO_2$, and the medium was changed every 48 h.

**3T3-L1 cell viability assay.** 3T3-L1 preadipocytes were plated at $1.0 \times 10^4$ cells/well. The cells were treated with various concentrations (7.81–500 μg/mL) of *M. paniculata* ethanol extract and then incubated at 37°C in a humidified 5% $CO_2$ atmosphere for 24 h. The viability of 3T3-L1 preadipocyte cells was assessed using a previously described 3-[4,5- dimethylthiazol-2-yl]-2,5 diphenyl tetrazolium bromide (MTT) assay protocol with slight modifications [67]. After the cells were incubated with 0.5 mg/ml MTT for 4h, 10% sodium dodecyl sulfate (SDS) was added to each well, followed by overnight incubation. Cell viability was measured at 540 nm using a microplate reader. Non-toxic concentrations of the extract were determined based on the viability assay results and used in subsequent experiments.

**Coating the plates.** A 1.3% gelatin solution (Sigma Aldrich, Saint Louis, USA) was prepared in phosphate-buffered saline (PBS). Gelatin solution (2 ml) was transferred to each well of a 6-well cell culture plate, followed by incubation at 37°C for 1 h. After incubation, the plates were dried for 30 min and sterilized using an ultraviolet (UV) lamp for 15 min.

**Differentiation of 3T3-L1 preadipocyte cells and the effect of *M. paniculata* ethanol extract on 3T3-L1 adipogenesis.** 3T3-L1 preadipocyte cells were differentiated into mature adipocytes on a gelatin-coated 6-well plate using a previously described protocol with slight modifications [68]. Before 3T3-L1 preadipocytes were differentiated by induction agent, cells were incubated for 48h with various concentrations of *M. paniculata* ethanol extract (7.8, 15.625, 31.25, 62.5, and 125 μg/ml) or only the medium (for adipocyte control and preadipocyte control groups). Cells were induced with induction agent I (dexamethasone 0.25 μm, insulin 2 μg/mL, isobutyl-methylxanthine (IBMX) 0.5 mM, and Rosiglitazone 2 μM in DMEM supplemented by 10% fetal bovine serum (FBS) for 48 h at 37°C in a humidified 5% $CO_2$ atmosphere, or only the medium (for preadipocytes control group). The medium was replaced, and the cells were treated with induction agent II (insulin 2 μg/mL in DMEM supplemented with 10% FBS) and incubated for 48 h. The medium was replaced every 2 d until droplets of lipids appeared at 70% confluency. These cells were used for further experiments.

**Quantitative polymerase chain reaction (qPCR).** Total RNA was extracted from each group using Genezol, according to the manufacturer's instructions (Geneaid Biotech Ltd., New Taipei, Taiwan). The mRNA levels of PPARG and EP300 were determined using quantitative polymerase chain reaction (qPCR) with a Sensifast SYBR No-ROX One-Step Kit Master Mix Kit (Bioline, London, United Kingdom) and analyzed using an Eco™ Real-Time PCR instrument (Illumina Inc., San Diego, USA). The following primers were used for the qPCR: PPARG, 5′ CCATTCACAAGAGCTGACCC 3′ (forward) and 5′ CCATAGTGGAAGCCTG ATGC 3′ (reverse); EP300, 5′ CTGATCCAGAGAAGCGCAAGC3′ (forward) and 5′ GCAG GATTTGCCTGACTGGC 3′ (reverse); GAPDH 5′ –CGTCTTCACCACCATGGAGA 3′

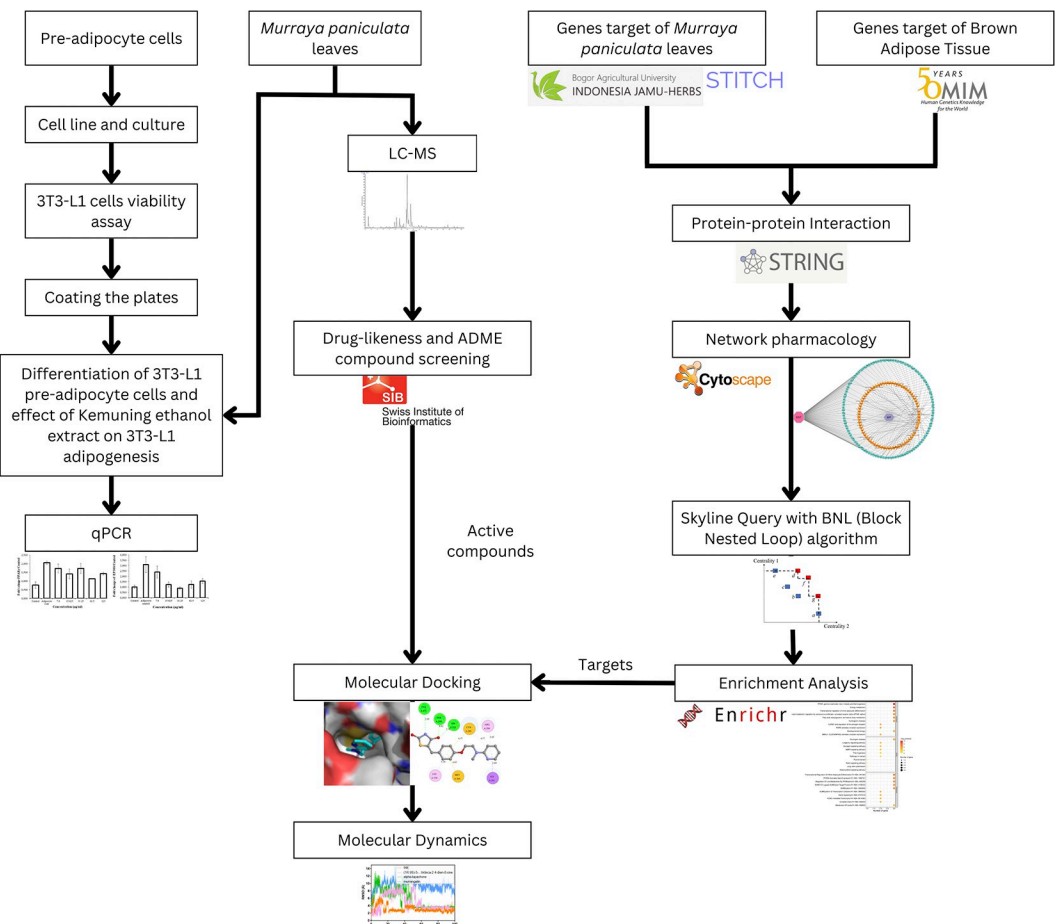

**Fig 3. The flow diagram of drug discovery research for obesity based on the integration of metabolite profiling, network pharmacology, molecular docking, molecular dynamics, and in-vitro tests.**

(forward) and 5′–CGGCCATCACGCCACAGTTT –3′ (reverse). The qPCR reaction cycling parameters were as follows: 45˚C for 10 min for reverse transcriptase activation, followed by a pre-incubation step at 95˚C for 2 min and amplification for 40 cycles at 95˚C for 5 sec, 55˚C for 10 sec and 72˚C for 5 sec. The relative expression ΔΔCq method was used to quantify gene expression, and GAPDH was used as a normalizer. All the samples were measured in duplicate.

**Statistical analysis.** The results were presented as mean ± standard deviation (SD). Statistical analysis was conducted using unpaired Student's t-test and analysis of variance (ANOVA) followed by a post-hoc test. Statistical significance was set at $p < 0.05$.

## Results

This study aimed to gain insights into the targets of obesity and the mechanisms of action of active compounds in *M. paniculata* in response to obesity (**Fig 3**).

### Network pharmacology

The targets obtained from OMIM were 114, whereas those from STITCH were 64 targets with 152 interactions, which were derived from the 76 active compounds of *M. paniculata* obtained

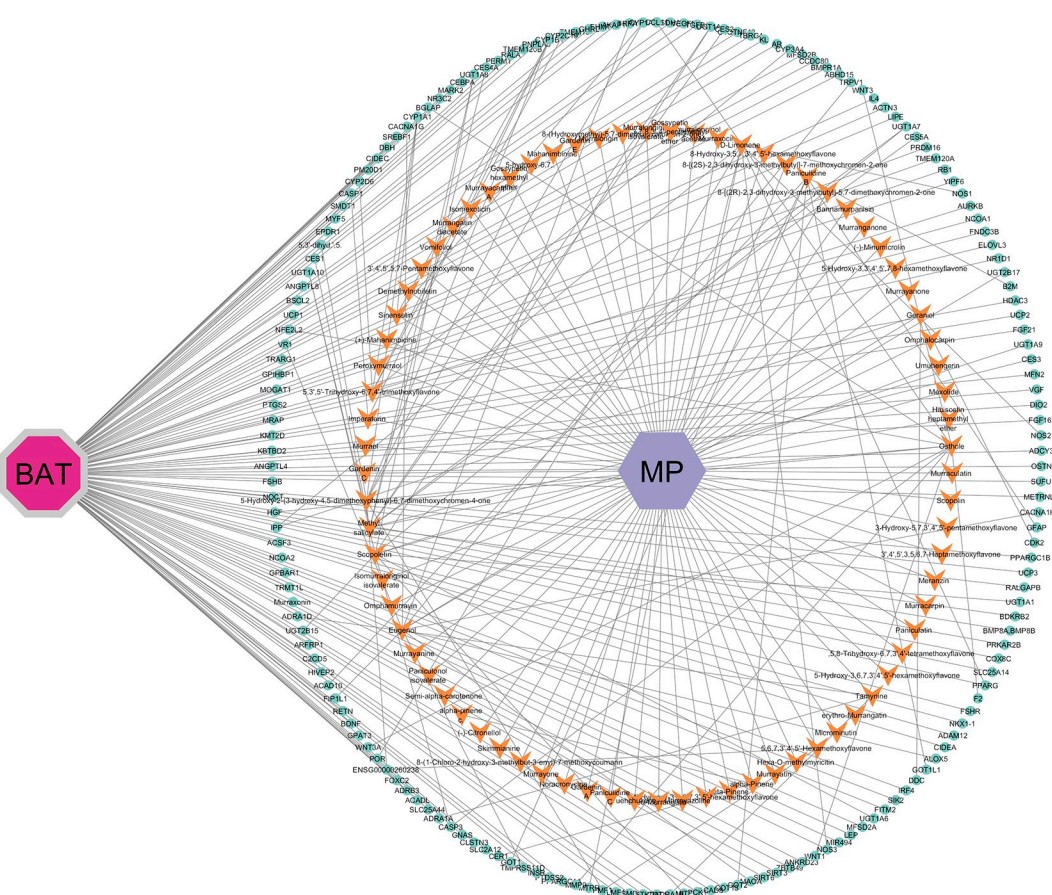

**Fig 4. Pharmacological network analysis of obesity–active compound *M. paniculata*–target proteins.** The pink octagon represents obesity (brown adipose tissue), the purple hexagon represents the herbal plant *M. paniculata*, the orange "V" sign represents the active compounds of *M. paniculata*, and the green circle represents the target proteins.

from IJAH analytics (**S9 Table**). Protein interactions related to obesity, after expansion, included 952 proteins with 3416 interactions, whereas those related to the active compound, *M. paniculata*, after expansion, included 468 proteins with 2336 interactions. The resulting protein interactions were then combined, and duplicates were removed. The elimination of the duplicates resulted in 1332 proteins with 5196 interactions. Network pharmacology was visualized using Cytoscape software. An illustration of the relationship between obesity, the active compound *M. paniculata* from IJAH analytics, and the target proteins is shown in **Fig 4**.

## Selecting protein target using skyline query

Network analysis in the form of centrality measures was performed on the target proteins, consisting of degree, betweenness, closeness, bridging, radiality, and eigenvector centralities. In this process, an undirected graph was used. In this study, we used the skyline query technique to obtain a set of dominant proteins from the graph.

The results of protein data processing using a skyline query showed that the three most significant proteins were EP300, PPARG, and peroxisome proliferator-activated receptor gamma coactivator 1-alpha (PPARGC1A), as shown in **Table 1**.

**Table 1. The results of the skyline query to process the potential target proteins.**

| No | Target | Degree | Betweenness | Closeness | Bridging | Radiality | Eigenvector |
|----|--------|--------|-------------|-----------|----------|-----------|-------------|
| 1 | EP300 | 64 | 162803.27192 | 2.602E-04 | 2742104 | 12.497721 | 0.23401973 |
| 2 | PPARGC1A | 65 | 146262.13907 | 2.626E-04 | 3402158 | 12.528714 | 0.29980809 |
| 3 | PPARG | 72 | 161827.65271 | 2.657E-04 | 1931188 | 12.569735 | 0.31835813 |

## Enrichment analysis

Enrichment analyses of EP300, PPARGC1A, and PPARG were performed using the enrichment database. Gene Ontology (GO) data from the three targets included 257 biological processes, three cellular components, and 45 molecular functions. Based on GO biological process data, three proteins were found to play a role in fat cell differentiation, EP300 and PPARGC1A regulate gluconeogenesis, and PPARG and PPARGC1A play a role in glucose homeostasis. PPARG also plays a role in regulating adiponectin secretion, positive regulation of adipose tissue development, and regulation of adipose tissue development. The results of the cellular components showed the presence of the three proteins in the nucleus and intracellular membrane-bound organelles. Molecular function analysis showed that these three proteins are related to RNA polymerase II-specific DNA-binding transcription factor binding, DNA-binding transcription factor binding, and DNA binding. The top 10 GOs with the lowest p-values are presented in **Fig 5**, and the overall results are shown in **S3**–**S5 Tables**.

The pathways for the three targets were as follows: 111 BioPlanet, 39 KEGG, and 126 Reactome. Based on the pathway from BioPlanet, the three targets were found to be involved in the

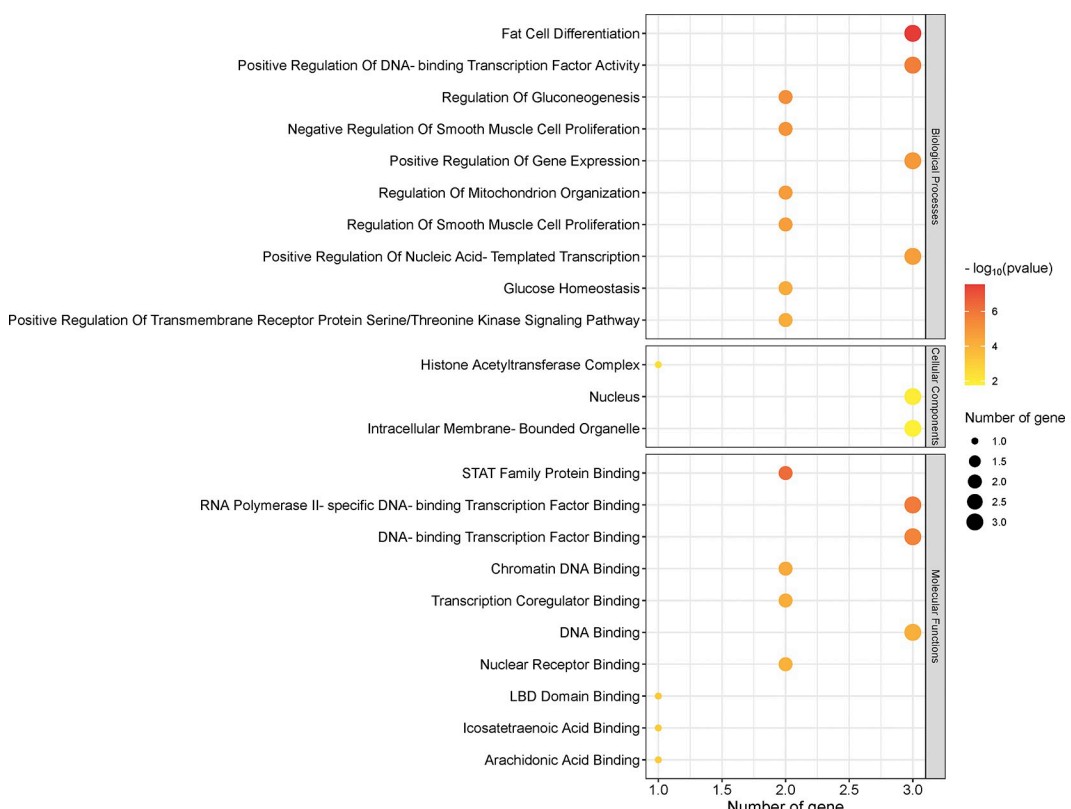

**Fig 5. Top 10 gene ontologies with the lowest p-value.**

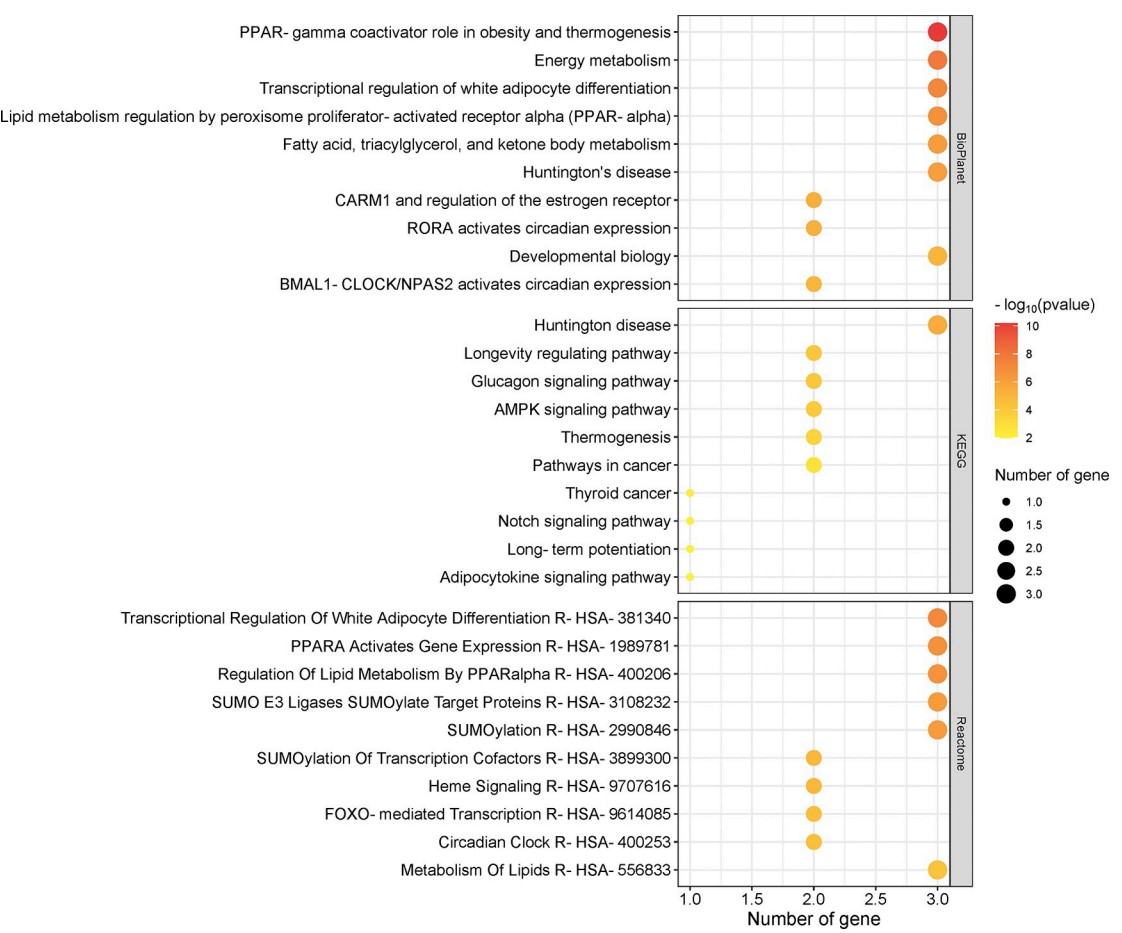

**Fig 6. Top 10 pathways with the lowest p-value.**

PPAR-gamma coactivator role in obesity and thermogenesis; energy metabolism; transcriptional regulation of white adipocyte differentiation; lipid metabolism regulation by peroxisome proliferator-activated receptor alpha (PPAR-alpha); fatty acid, triacylglycerol, and ketone body metabolism; Huntington's disease; CARM1 and regulation of the estrogen receptor; RORA activation of circadian expression; and developmental biology. PPARG and PPARGC1A are involved in adipogenesis. KEGG analysis showed that PPARG and PPARGC1A were related to the longevity regulating pathway, glucagon signaling pathway, AMPK signaling pathway, and thermogenesis. EP300 and PPARGC1A are related to the glucagon signaling pathway, and PPARGC1A is related to the adipocytokine signaling pathway. In the Reactome pathway, three proteins were found to be involved in the transcriptional regulation of white adipocyte differentiation: PPARA activates gene expression, regulation of lipid metabolism by PPARalpha, SUMO E3 ligases SUMOylate target proteins, SUMOylation, and metabolism of lipids. The top ten pathways with the lowest p-values are presented in **Fig 6**, and the overall results are shown in **S6**–**S8 Tables**.

## Metabolite profiling

Using sonication, *M. paniculata* leaf samples were extracted with three different solvents: water, 50% ethanol, and ethanol. This method was chosen because it can increase the yield and quality of bioactive compounds contained in plants. Furthermore, extraction using this

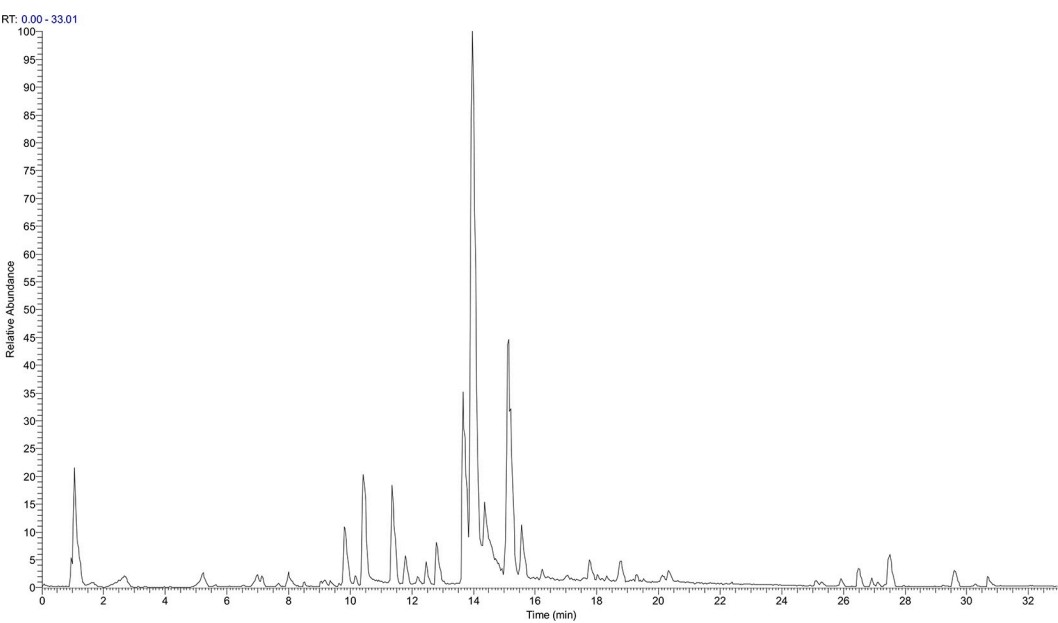

**Fig 7. The base peak chromatogram in positive ionization mode for the ethanol extract of *M. paniculata*.**

method can be performed at low temperatures in a short period, thereby minimizing the use of solvents. In this method, cavitation occurs by breaking down cell walls, thus expanding the solvent contact and yield [69]. Notably, polar and semi-polar solvents were used in this study. Ethanol was selected because it is commonly used for herbal plant extraction. The base peak chromatogram in positive ionization mode for the ethanol extract is shown in **Fig 7**. LC-MS results from *M. paniculata* leaf extract predicted 16 active compounds. These compounds are listed in **S1 Table**.

## Drug-likeness of the compounds

ADME parameters were used to determine the drug-likeness of *M. paniculata* leaf metabolites. The measurement results are listed in **S2 Table**. Lipinski's rule of five is a physicochemical parameter consisting of molecular weight < 500 D, logP < 5, hydrogen bond donors < 5, and hydrogen bond acceptors < 10. This parameter refers to active compounds administered orally and is related to their solubility in water and permeability in the intestine [50]. The molar refractivity of the compounds was calculated, and values between 40–130 are considered ideal [70]. The selected compound must also have a high gastrointestinal absorption and bio-availability score of > 0.3 (30%) [51]. Based on the calculation results (**S2 Table**), only ten compounds were docked: (1R,9S)-5-[(E)-2-(4-Chlorophenyl)vinyl]-11-(5-pyrimidinylcarbo-nyl)-7,11-diazatricyclo[7.3.1.02,7]trideca-2,4-dien-6-one, alpha-lapachone, trans-3-indo-leacrylic acid, DL-tryptophan, Hainanmurpanin, Murraol, Murralongin, Murrangatin, L-phenylalanine, and 4-aminobenzoic acid.

## Molecular docking

Ten compounds from the drug-likeness calculations and two target proteins (PPARG and EP300) were selected for molecular docking using AutoDock Vina. Redocking was performed between the proteins and native ligands in the crystal structure. The binding affinity result was -8.8 kcal/mol for PPARG and its native ligand BRL, while that between EP300 and protein 99E

**Table 2. The binding affinity results of the ligands and protein targets.**

| No | Compound | Binding Affinity (kcal/mol) | |
|---|---|---|---|
| | | **PPARG** | **EP300** |
| 1 | (1R,9S)-5-[(E)-2-(4-Chlorophenyl)vinyl]-11-(5-pyrimidinylcarbonyl)-7,11-diazatricyclo[7.3.1.02,7]trideca-2,4-dien-6-one | -8.8 | -8.7 |
| 2 | alpha-lapachone | -7.8 | -7.9 |
| 3 | trans-3-indoleacrylic acid | -6.9 | -6.6 |
| 4 | DL-tryptophan | -6.7 | -5.9 |
| 5 | Hainanmurpanin | -7 | -5.8 |
| 6 | Murraol | -6.7 | -6.1 |
| 7 | Murralongin | -6.4 | -6.4 |
| 8 | Murrangatin | -6.4 | -6.4 |
| 9 | L-phenylalanine | -6.3 | -6.1 |
| 10 | 4-aminobenzoic acid | -5.2 | -5.6 |

was -8.4 kcal/mol. The grid box used for redocking was the benchmark for the docking compounds from *M. paniculata* leaves.

Molecular docking results showed that all binding affinities obtained were ≤ -5.0 kcal/mol, indicating that the ligands strongly bind with protein targets [71]. The best binding results for both PPARG and EP300 proteins were obtained with the ligands (1R,9S)-5-[(E)-2-(4-Chlorophenyl)vinyl]-11-(5-pyrimidinylcarbonyl)-7,11-diazatricyclo[7.3.1.02,7]trideca-2,4-dien-6-one, alpha-lapachone, and trans-3-indoleacrylic acid. Binding with (1R,9S)-5-[(E)-2-(4-Chlorophenyl)vinyl]-11-(5-pyrimidinylcarbonyl)-7,11-diazatricyclo[7.3.1.02,7]trideca-2,4-dien-6-one produced the highest binding affinities of -8.8 and -8.7 for PPARG and EP300, respectively, followed by alpha-lapachone with a binding affinity of -7.8 and -7.9 for PPARG and EP300, respectively, and finally trans-3-indoleacrylic acid with -6.9 and -6.6 for PPARG and EP300, respectively. The binding-affinity results are presented in **Table 2**. The 2D and 3D visualization results between proteins and all compounds are shown in the **S1 File** (PPARG) and **S2 File** (EP300).

## Molecular dynamics simulations

In MD simulations, RMSD and RMSF calculations are commonly employed to assess stability and conformational changes in ligands and proteins [72]. In this study, the top 10 ligands identified from the docking studies underwent 100-ns MD simulations to elucidate their binding interactions and refine the binding poses acquired from the initial docking studies. Among the ligands bound to PPARG, trans-3-Indoleacrylic acid exhibited the most stable behavior, with an average RMSD of 0.99 Å (**Fig 8A**). However, it is noteworthy that despite its overall stability, trans-3-Indoleacrylic acid experienced a transient increase in RMSD, averaging 2.14 Å between 70.62 ns and 75.70 ns. The next two ligands with the lowest average RMSD values were 4-aminobenzoic acid and (1R,9S)-5-[(E)-2-(4-chlorophenyl)ethenyl]-11-(pyrimidine-5-carbonyl)-7,11-diazatricyclo[7.3.1.02,7]trideca-2,4-dien-6-one, with values of 1.94 Å and 2.83 Å, respectively. Remarkably, the average RMSD values of trans-3-Indoleacrylic acid and 4-aminobenzoic acid are even lower than that of the native ligand, BRL, which recorded a value of 2.02 Å, indicating fewer changes in the overall structure during the simulation period for both ligands. Unlike trans-3-Indoleacrylic acid, there were no significant spikes in the RMSD values of 4-aminobenzoic acid or (1R,9S)-5-[(E)-2-(4-chlorophenyl)ethenyl]-11-(pyrimidine-5-carbonyl)-7,11-diazatricyclo[7.3.1.02,7]trideca-2,4-dien-6-one. In the EP300-ligands complexes, convergence of the RMSD was observed after 60 ns, as the RMSD values

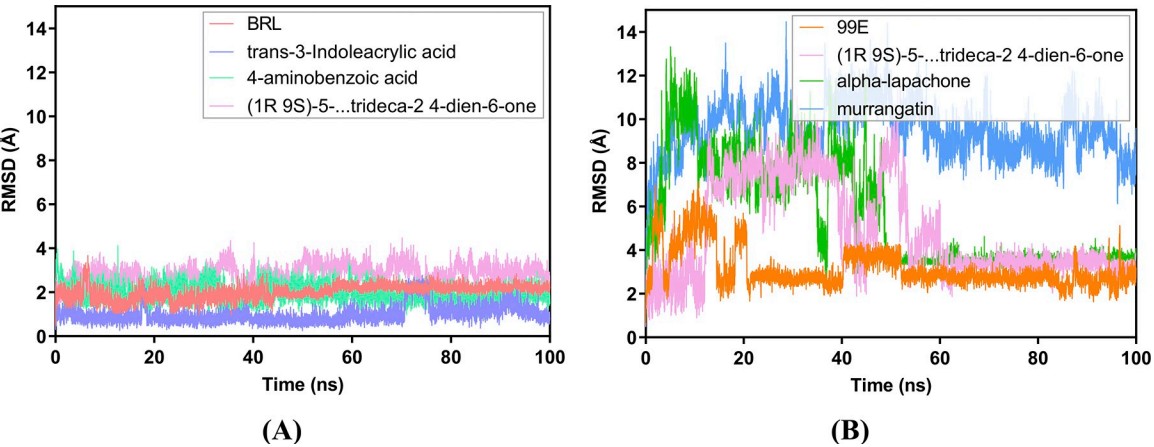

**Fig 8.** RMSD of the three best ligands compared to native ligands in ligand-PPARG (**A**) and ligand-EP300 (**B**) structures.

exhibited significant fluctuations beforehand. (1R,9S)-5-[(E)-2-(4-chlorophenyl)ethenyl]-11-(pyrimidine-5-carbonyl)-7,11-diazatricyclo[7.3.1.02,7]trideca-2,4-dien-6-one showed the most stable simulation system, having the lowest average RMSD value of 3.33 Å, slightly lower than the second most stable ligand, alpha-lapachone with 3.57 Å (**Fig 8B**). However, both ligands still had lower average RMSD values than native ligand 99E, which was 2.70 Å. The next ligand with the lowest average RMSD value was murrangatin, but the difference between the two preceding ligands was significant, with a value of 8.94 Å. Collective RMSD graphs of all ligands compared with the native ligands can be observed in the **S3 File** (PPARG) and **S4 File** (EP300).

RMSF analysis was conducted for both PPARG and EP300. As shown in **Fig 9A**, the RMSF plot for BRL exhibited higher average values than the other top three ligands, averaging 0.305 Å, while (1R,9S)-5-[(E)-2-(4-Chlorophenyl)vinyl]-11-(5-pyrimidinylcarbonyl)-7,11-diazatricyclo[7.3.1.02,7]trideca-2,4-dien-6-one, 4-aminobenzoic acid, and trans-3-indoleacrylic acid averaged 0.165 Å, 0.224 Å, and 0.179 Å, respectively. This suggests that all the standard ligands

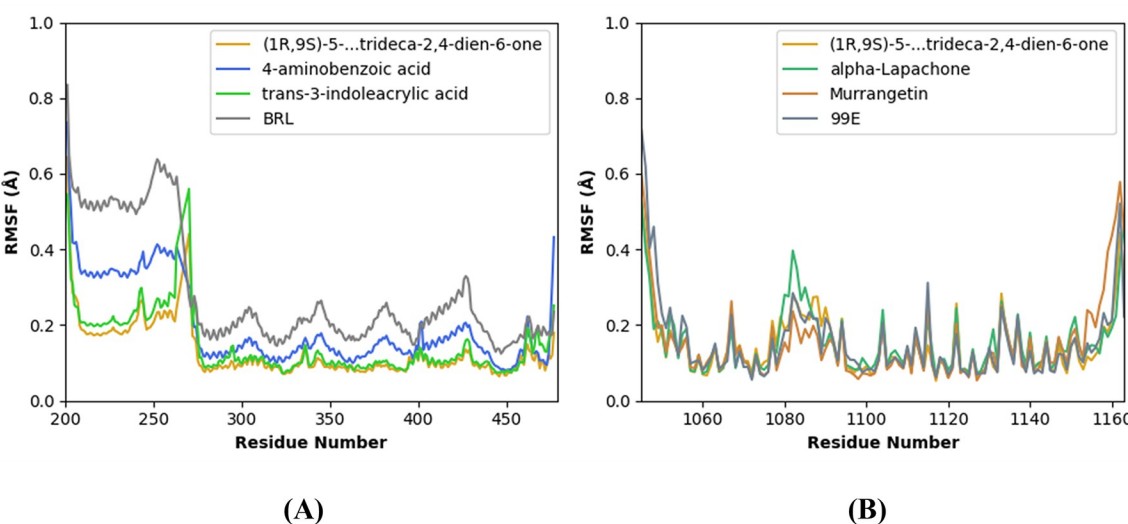

**Fig 9.** RMSF of PPARG (a) and EP300 (b) in the three best ligands and native ligand-bound structures.

demonstrated greater stability during complex formation than the PPARG-BRL complex. However, the RMSF of both PPARG and the complex comprising 4-aminobenzoic acid and (1R,9S)-5-[(E)-2-(4-chlorophenyl)ethenyl]-11-(pyrimidine-5-carbonyl)-7,11-diazatricyclo[7.3.1.02,7] trideca-2,4-dien-6-one increased significantly, particularly at the Leu270-Gln271 residues. This is because the side chain of Gln271 tended to rotate, transitioning from facing away from the ligand to moving closer and establishing better interactions with the ligand. This phenomenon enhances ligand stability by effectively closing the binding pocket. Unlike PPARG, the disparity in the average RMSF for the top three ligands was not significantly different from that for the 99E ligand (**Fig 9B**). The average RMSF for (1R,9S)-5-[(E)-2-(4-chlorophenyl)ethenyl]-11-(pyrimidine-5-carbonyl)-7,11-diazatricyclo[7.3.1.02,7]trideca-2,4-dien-6-one, alpha-lapachone, and murrangetin was 0.147 Å, 0.156 Å, and 0.154 Å, respectively, compared to the 99E ligand (0.155 Å). Additionally, the peak surge at Gln1082 for the EP300-ligand complexes was triggered by the presence of the ligand at its binding site, which is in close proximity to the aforementioned residue. Consequently, in the EP300-alpha-lapachone complex, this peak was observed because of the loss of the alpha helix structure at that residue, leading to its destabilization and an increase in its RMSF. The RMSF graphs of PPARG and EP300 for all ligands and native ligand-bound structures are shown in the **S5** and **S6 Files**.

## Viability assay of the *M. paniculata* extract on 3T3-L1 preadipocyte cells

The cytotoxicity of *M. paniculata* ethanolic extract was evaluated at various concentrations to determine its safe concentrations. 3T3-L1 preadipocyte cells treated with extract at concentrations up to 125 μg/ml showed no differences in the percentage of cell viability compared to untreated cells (**Fig 10**), suggesting that *M. paniculata* extract is safe for 3T3-L1 preadipocyte cells up to a concentration of 125 μg/ml.

## *M. paniculata* ethanolic extract decreased adipocyte differentiation in 3T3-L1 adipocyte cells

Adipogenesis leads to the accumulation of excess fat in adipocytes during the adipocyte differentiation process [73]. Several important factors involving the cascade of adipocyte

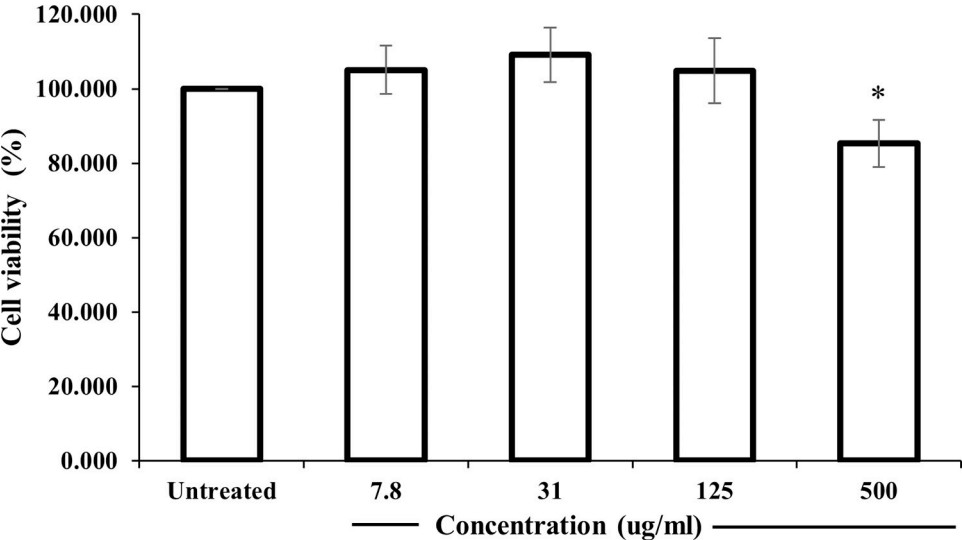

**Fig 10. 3T3-L1 cell viability after treatment with *M. paniculata* ethanolic extract.** Assays were conducted in three-independent experiments. Values are expressed as means ± SD. *p < 0.05, vs. Untreated.

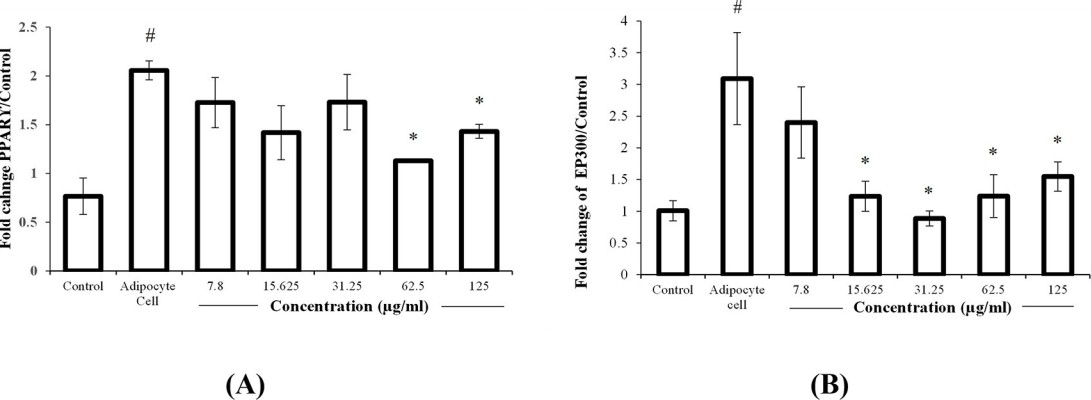

**Fig 11. Adipocyte differentiation in 3T3-L1 cells after treatment with *M. paniculata* ethanolic extract for 48h at various concentrations.** (**A**) PPARG mRNA and (**B**) EP300 mRNA expression levels. GAPDH was used as a normalizer. Values are the mean ± standard deviation (SD). #p < 0.05, vs. Control; *p < 0.05 vs adipocyte control.

differentiation have been discovered, including the transcription factors CCAAT/enhancer binding protein alpha (C/EBPα) and peroxisome proliferator-activated receptor gamma (PPARG) [74]. Activation of PPARG need coactivator recruitment including CREB-binding protein (CBP), histone acetyltransferas p300 (EP300) and PPAR-binding protein (PBO) to initiates adipogenesis programs [75]. In this study, we determined the gene expression of PPARG and EP300 in 3T3-L1 adipocytes after treatment with *M. paniculata* ethanolic extracts for 48 h to explore the effect of the extract on 3T3-L1 differentiation. **Fig 11A and 11B** show that adipocyte control cells expressed higher levels of PPARG and EP300 mRNA than the control cells significantly, suggesting that adding an induction agent stimulated the differentiation of 3T3-L1 preadipocytes into mature adipocytes. Treated 3T3-L1 mature adipocytes with *M. paniculata* ethanolic extract for 48 h showed reduced gene expression of PPARG starting at a concentration of 62.5 μg/ml relative to adipocyte cells (**Fig 11A**). In addition, *M. paniculata* ethanolic extract also decreased EP300 mRNA expression in 3T3-L1 adipocytes significantly compared to adipocyte cells at all concentration extracts except 7.8 μg/ml as shown in **Fig 11B**. This result suggested that *M. paniculata* ethanolic extracts could reduce differentiation of preadipocyte 3T3-L1 into mature adipocyte after incubation for 48h as indicated by the mRNA expression of PPARG and EP300 in 3T3-L1 mature adipocyte.

## Discussion

Research into herbal medicines has increased in recent years. Various approaches have been used to uncover herbal potential, including the network pharmacology approach. Network pharmacology predicts the interactions between target proteins and drugs and their relationship with diseases [23, 76–79]. This study used a network pharmacology approach with the skyline query technique and the block-nested loop (BNL) algorithm to obtain the dominant protein in the protein network from obesity and *M. paniculata* compounds. Network pharmacology research with skyline queries has been conducted, but it uses the top-k algorithm [80, 81], therefore, BNL is a new approach. The results of skyline query analysis showed that EP300, PPARG, and PPARGC1A were dominant and, therefore, have potential as targets for treating obesity. Enrichment analysis was performed to determine the functionality of these potential targets in the form of gene ontologies and pathways. Gene ontologies show target roles in fat cell differentiation, glucose homeostasis, and regulation of adipose tissue

development. These pathways are potential targets for obesity, thermogenesis, energy metabolism, and adipocytokine signaling.

Metabolite profiling using LC-MS was performed to obtain the active compounds from *M. paniculata* leaves. Of the sixteen potentially active compounds, ten were filtered and selected based on their drug-likeness and ADME. The ten filtered compounds and their target proteins (PPARG and EP300) were simulated using molecular docking and molecular dynamics simulations to identify the bonding between the compounds and proteins.

Molecular dynamics simulations revealed that the best RMSD values with PPARG were for trans-3-indoleacrylic acid, 4-aminobenzoic acid, and (1R,9S)-5-[(E)-2-(4-Chlorophenyl) vinyl]-11-(5-pyrimidinylcarbonyl)-7,11-diazatricyclo[7.3.1.02,7]trideca-2,4-dien-6-one, while (1R,9S)-5-[(E)-2-(4-Chlorophenyl)vinyl]-11-(5-pyrimidinylcarbonyl)-7,11-diazatricyclo [7.3.1.02,7]trideca-2,4-dien-6-one, alpha-lapachone, and murrangatin were the EP300-ligands with the best RMSD value. The binding visualization results of the molecular docking between proteins and ligands are shown in **S1** and **S2 Files**.

We also investigated the inhibitory effect of *M. paniculata* extract on adipocyte differentiation in preadipocyte 3T3-L1 cells. Lipid formation and adipogenesis were determined based on the expression of adipogenesis-inducing genes PPARG and EP300. Cells were induced with a cocktail reagent for 12 days, and the expression of PPARG and EP300 mRNA was significantly enhanced relative to that in undifferentiated 3T3-L1 cells. Preadipocyte 3T3-L1 cells treated with *M. paniculata* ethanolic extract at 62.5 and 125 μg/ml for 48 h decreased the PPARG gene expression significantly compared to adipocyte cells. On the other hand, *M. paniculata* ethanolic extract at a concentration of 15.625 μg/ml reduced the expression of EP300 genes, meaning that *M. paniculata* ethanolic extract treatment prevented the lipid formation or adipogenesis in pre-adipocyte by downregulating the expression of PPARG and EP300 genes in mature 3T3-L1 adipocytes. These results suggest that the treatment of *M. paniculata* extract inhibited adipocyte maturation and enhanced intracellular lipid metabolism by reducing the expression of genes involved in lipogenesis. This process led to decreased lipid accumulation and formation, ultimately causing anti-obesity effects.

Obesity is an adipose tissue dysfunction and can disrupt health [5, 7, 82, 83]. This has caused obesity to become an epidemic [5], which has worsened over the last 50 years [82]. Causes of obesity include genetics, lack of physical activity, insomnia, endocrine disorders, medications, excess carbohydrates, and decreased energy metabolism [82]. Obesity caused diabetes mellitus [7, 84, 85], hypertension [86–88], cardiovascular disease [83, 89–91], and dyslipidemia [92, 93].

The type of adipose tissue that is often associated with obesity is WAT. WAT is classified into two main depots: visceral adipose tissue (VAT) and subcutaneous adipose tissue (SAT). WAT is associated with metabolic and endocrine functions and plays a role in regulating energy homeostasis and insulin sensitivity [94]. Dysfunction of WAT causes obesity and contributes to the development of metabolic dysregulation [95, 96]. Dysfunction occurs due to an imbalance between energy intake and expenditure [97, 98] which is characterized by loss of adipose tissue expansion capacity, increased adipocyte hypertrophy, and changes in the secretory profile of adipose cells (macrophage accumulation and inflammation) [95, 96]. In this study, 3T3-L1 pre-adipocyte cells were used. Basal bioenergetic and gene expression profiles of 3T3-L1 adipocytes showed typical WAT. Differentiation of 3T3-L1 into mature adipocyte is a model of white adipocytes formation that is often used for anti-obesity in vitro studies [99].

Research on inhibiting adipogenesis from 3T3-L1 cells has been carried out using olvanil. PPARG, PREF1, and FABP4 are used as markers related to adipocyte maturation. The results showed a decrease in PPARG and FABP4 gene expression, while there was an increase in PREF1 gene expression. The in vitro study results demonstrated the olvanil effect inhibiting

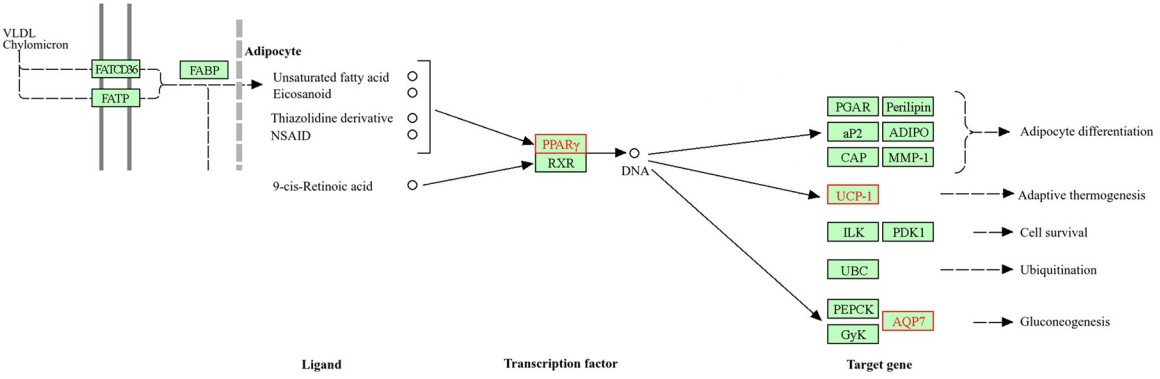

**Fig 12. PPARG signaling pathway based on KEGG pathway [106].**

adipogenesis, reducing lipid accumulation and triglycerides in 3T3-L1 adipocytes [100]. This is in accordance with the research carried out, namely that there was downregulated PPARG expression in 3T3-L1 adipocytes.

To understand the role of E300 in adipogenesis, a study was carried out observing the pattern of histone butyrylation in 3T3L1 cells during adipogenesis. EP300-catalyzed increases in histone butyrylation were found during adipogenesis, so EP300-specific inhibition of butyrylation may lead to repression of adipogenesis [101]. The results of this study explain that the reduction in EP300 gene expression can be targeted to inhibit adipogenesis and be used to treat obesity.

PPARG is a ligand-dependent transcription factor that plays a role in adipogenesis and insulin sensitivity [102–104]. PPARG is a key factor in regulating the differentiation of WAT and BAT. PPARG activated and bind to retinoid X receptor (RXR) to form PPARG-RXR heterodimer [105]. PPARG-RXR heterodimers are useful in adipocyte differentiation and adaptive thermogenesis (**Fig 12**). Dysregulation of PPARG has the potential to cause obesity due to its important role in adipogenesis and adipocyte gene expression. Research on PPARG expression shows that PPARG mRNA expression is most abundant in the serum of obese, diabetic, and non-diabetic patients. PPARG mRNA expression was positively correlated with body mass, waist circumference, and waist-to-hip ratio of obese patients. Treating obesity and type 2 diabetes with PPARG and RXR antagonists can reduce triglyceride content in white adipose tissue, increasing fatty acid burning and energy expenditure [105].

PPARG, activated by ligands, can induce the formation of brown (beige) fat genes in WAT. The formation of the brown fat gene is influenced by the activity of the histone-lysine N-methyltransferase PRDM16 pathway, which is a factor controlling the development of classical brown fat [107–109]. Brown and beige fat are adipose tissues that play a central role in the pathology of obesity, namely, in thermogenesis [110]. The process of heat production consumes a large amount of energy. Therefore, increasing BAT activity is an attractive strategy for fighting obesity. Previous research has shown that increased BAT activity improves the metabolic profile in obesity, whereas decreased BAT activity is associated with increased adiposity and the incidence of metabolic syndrome [111]. BAT and WAT differ in the number and density of mitochondria, number and size of lipid droplets, and expression levels of uncoupling protein-1 (UCP). Additionally, it plays a major role in thermogenesis (heat production). UCP-1 is mainly located in the inner mitochondria, where it converts chemical energy into heat energy. This functional property of the thermogenic adipocytes protects animals from hypothermia during hibernation [6].

Based on our results, the binding between PPARG and the native ligand occurs via a hydrogen bond at the amino acids Tyr473, Ser289, and His323; hydrophobic interactions at Arg288, Leu330, and Cys285; and pi-sulfur bonds at Cys285 and Met364. Previous research using the PPARG protein structure showed that the binding site between the PPARG protein and the ligand consists of a canonical orthosteric interaction with Tyr473, which is included in the activation function of helix 12, and an allosteric interaction with Arg288 [52]. Molecular docking of the active compound of *M. paniculata* showed hydrophobic interactions with Arg288 by several ligands, including (1R,9S)-5-[(E)-2-(4-chlorophenyl)ethenyl]-11-(pyrimidine-5-carbonyl)-7,11-diazatricyclo[7.3.1.02,7]trideca-2,4-dien-6-one, alpha-lapachone, DL-tryptophan, Hainanmurpanin, L-phenylalanine, Murrangatin, Murraol, and trans-3-indoleacrylic acid.

EP300 is a histone acetyltransferase (HAT) that encodes the proteins involved in transcriptional regulation [112]. EP300 influences the development of adipose tissue, promoting the expression of genes related to adipogenesis via histone acetylation [101]. EP300 is also a key mediator of cellular homeostasis [113], making it an essential therapeutic target for obesity. Obesity is a metabolic disorder affecting metabolic organs, including the liver, which regulates lipid and glucose homeostasis. Metabolic disorders are usually accompanied by an inflammatory response that can cause liver disease [114]. Based on a study conducted by Bricambert et al. (2010), EP300 is responsible for the development of hepatic steatosis in obesity and type 2 diabetes [115].

Histone acetyltransferases (HATs), also known as bromodomains, play a role in the acetylation of lysine residues in histone tails [116]. Two conserved amino acid residues in the binding pocket of the bromodomain, Asn and Tyr, interact to introduce lysine acetate [117]. The results of molecular docking showed that the binding of EP300 and the ligand consisted of hydrogen bonds (Pro1074 and Asn1132), hydrophobic interactions (Leu1084, Leu1083, Val1079, Val1138, and Val1128), and pi-donor hydrogen bonds (Asn1132 and Phe1075). Studies using the crystal structure of the protein have shown that XDM-CBP binds to the ligand at residues Arg1137, Pro1074, Asn 1132, Leu 1084, Val1079, Tyr1089, Gln1077, Phe1075, and Ile1086 [117]. Binding to the Asn1132 residue is observed in several active compounds, including (1R,9S)-5-[(E)-2-(4-chlorophenyl)ethenyl]-11-(pyrimidine-5-carbonyl)-7,11-diazatricyclo[7.3.1.02,7]trideca-2,4-dien-6-one, 4-aminobenzoic acid, DL-tryptophan, l-phenylalanine, murraol, and trans-3-indoleacrylic acid. Here, we used metabolite profiling, network pharmacology, and molecular docking approaches to target obesity-related proteins using *M. paniculata* active compounds.

Several studies have investigated the effects of EP300 on adipogenesis and lipid metabolism. For example, C646, a CBP/p300 inhibitor, has been shown to reduce adiposity in zebrafish larvae, block proliferation, cause cell cycle arrest, and stimulate differentiation of adipose-derived stem cells in goats [118]. A reduction in EP300 expression following the administration of C646 has been reported. We found that C646 suppressed IRS1/2 acetylation and triggered IRS1/2 membrane translocation, thus activating the insulin signaling pathway. C646 increases IR beta activity, binds IRS, and triggers IRS tyrosine phosphorylation. This regulation causes the activation of Pi3K-AKT signaling, suppresses glucose production in the liver, and improves hyperglycemia. These results demonstrate the importance of epigenetic factors in the pathology of metabolic disorders, including obesity and diabetes mellitus [119].

The results of this study also strengthen the findings regarding the role of epigenetics, especially EP300, in metabolic dysfunction related to adipogenesis. Various internal and external factors influence adipocyte function and differentiation. Individual genetic and environmental factors cause changes in the metabolism of the body, which in turn change the epigenetic profile, resulting in changes in gene expression and triggering obesity. Adipocyte function and

differentiation are influenced by CBP/p300, a super-enhancer epigenomic regulator that controls chromatin landscaping during this process [120].

A HAT inhibitor, curcumin (diferuloylmethane), from *Curcuma longa* roots, has been shown to specifically inhibit HAT activity in CBP/E300. Administration of curcumin to mice fed with a high-fat diet (HFD) and induced diabetes improved the glycemic status and insulin sensitivity, reduced NF-kB in the liver, reduced macrophage infiltration into WAT, and increased adiponectin production [121]. Long-term administration of curcumin also causes HFD-induced obesity and insulin resistance in mice by suppressing inflammatory responses due to adiposity and hepatic lipogenesis [122].

*M. paniculata* is an herbal plant with potential as an anti-obesity drug [23, 123, 124]. *M. paniculata* leaf extract can reduce the levels of glutamic oxaloacetic transaminase (SGOT) and glutamic pyruvic transaminase (SGPT) enzyme activity in obese patients [125]. Infusion of *M. paniculata* leaves for 15 days also reduced the body mass index (BMI) of obese patients [14].

Five coumarins in *M. paniculata*, namely, mexoticin (1), omphalocarpin (2), murrangatin (3), kimcuongin (4), and murracarpin (5), were isolated and characterized. Their structures were determined using ESI-MS, HR-ESI-MS, and NMR (1D and 2D). The inhibitory effects of the coumarins on soluble epoxide hydrolases were also investigated. Among them, coumarins (2–4) exhibited inhibitory activity against soluble epoxide hydrolase, with $IC_{50}$ values of $2.2 \pm 4.7$, $13.9 \pm 6.5$, and $3.2 \pm 4.5$ μM, respectively. Kinetic analysis of the five coumarins revealed a noncompetitive enzymatic mode for compounds 3 and 4 and a combination of competitive and noncompetitive enzymatic modes for coumarin 2. Molecular modeling studies indicated that coumarin kimcuongin (4) demonstrated the most favorable binding profile within the active sites of human soluble epoxide hydrolases [124]. Cholesterol-related disorders have been linked to soluble epoxide hydrolases that play a significant role in the metabolism of cholesterol precursor metabolism [123]. In the present study, murrangatin was detected in the leaves of *M. paniculata*.

Apart from murrangatin, trans-3-indoleacrylic acid is a potential compound from *M. paniculata* with low binding affinity and RMSD values, as shown in the simulation results (**Table 2** and **Fig 8**). Trans-3-indoleacrylic acid is a metabolite of tryptophan secreted by gut microbiota [126, 127]. Studies have reported that trans-3-indoleacrylic acid (indole-3-acrylic acid) levels are negatively correlated with body weight in mice supplemented with inulin by regulating the gut microbiota [127]. In addition, trans-3-indoleacrylic acid enhances anti-inflammatory responses [126]. Obesity is associated with inflammation, and gut microbial dysbiosis is closely related to obesity [128–130].

## Conclusion

In this study, we demonstrated the potential of *Murraya paniculata* in the treatment and management of obesity based on network pharmacology using the skyline query technique and the block-nested loop (BNL) algorithm. We integrated network pharmacology, metabolite profiling, molecular docking, molecular dynamics simulations, and in vitro tests to identify the possible mechanisms of action of these compounds. Integration analysis using a skyline query revealed three main targets, EP300, PPARG, and PPARGC1A, which are potential targets for fighting obesity. We revealed ten potential compounds of *M. paniculata* and performed molecular docking and molecular dynamics simulations. The results of our molecular docking study showed good binding between proteins and ligands, and molecular dynamics evidence showed the stability of trans-3-indoleacrylic acid, 4-aminobenzoic acid, and (1R,9S)-5-[(E)-2-(4-chlorophenyl)ethenyl]-11-(pyrimidine-5-carbonyl)-7,11-diazatricyclo[7.3.1.02,7]trideca-2,4-dien-6-one for PPARG, and (1R,9S)-5- [E-2-(4-chlorophenyl)ethenyl]-11-(pyrimidine-5-carbonyl)-

7,11-diazatricyclo[7.3.1.02,7]trideca-2,4- dien-6-one, alpha-lapachone, and murrangatin for EP300. The in vitro tests results confirmed that *M. paniculata* induced decrease in PPARG and EP300 mRNA expression in 3T3-L1 preadipocytes. This demonstrates the potential of *M. paniculata* to be developed as an alternative therapy for obesity. In addition, to our study showing the potency of *M. paniculata* in treating obesity, further research is needed to confirm the results using the standard analog drugs PPARG and EP300 and to measure lipid accumulation and expression in brown adipose tissue via in vitro assay.

## Supporting information

**S1 Table. Prediction results of active compounds in *M. paniculata* leaves using LC-MS.**
(PDF)

**S2 Table. Results of drug-likeness of the compounds using SwissADME.**
(PDF)

**S3 Table. Gene ontology: Biological processes of PPARG, EP300, ad PPARGC1A.**
(PDF)

**S4 Table. Gene ontology: Cellular components of PPARG, EP300, ad PPARGC1A.**
(PDF)

**S5 Table. Gene ontology: Molecular functions of PPARG, EP300, ad PPARGC1A.**
(PDF)

**S6 Table. BioPlanet pathway of PPARG, EP300, ad PPARGC1A.**
(PDF)

**S7 Table. KEGG pathway of PPARG, EP300, ad PPARGC1A.**
(PDF)

**S8 Table. Reactome pathway of PPARG, EP300, ad PPARGC1A.**
(PDF)

**S9 Table. Dataset of targets from OMIM and STITCH, and *Murraya paniculata* compounds from IJAH analytics.**
(PDF)

**S1 File. Binding results of molecular docking between PPARG (7AWC) and ligands.** (A) Native ligand BRL, (B) (1R,9S)-5-[(E)-2-(4-chlorophenyl)ethenyl]-11-(pyrimidine-5-carbonyl)-7,11-diazatricyclo [7.3.1.02,7]trideca-2,4-dien-6-one, (C) 4-Aminobenzoic acid, (D) alpha-Lapachone, (E) DL-Tryptophan, (F) Hainanmurpanin, (G) L-Phenylalanine, (H) Murralongin, (I) Murrangatin, (J) Murraol, and (K) trans-3-Indoleacrylic acid.
(PDF)

**S2 File. Binding results of molecular docking between EP300 (5NU5) and ligands.** (A) Native ligand 99E, (B) (1R,9S)-5-[(E)-2-(4-chlorophenyl)ethenyl]-11-(pyrimidine-5-carbonyl)-7,11-diazatricyclo [7.3.1.02,7]trideca-2,4-dien-6-one, (C) 4-Aminobenzoic acid, (D) alpha-Lapachone, (E) DL-Tryptophan, (F) Hainanmurpanin, (G) L-Phenylalanine, (H) Murralongin, (I) Murrangatin, (J) Murraol, and (K) trans-3-Indoleacrylic acid.
(PDF)

**S3 File. The RMSD graphs of all ligands compared to the native ligands of PPARG.**
(PDF)

**S4 File. The RMSD graphs of all ligands compared to the native ligands of EP300.**
(PDF)

**S5 File. The RMSF graphs of PPARG in all ligand and native ligand-bound structures.**
(PDF)

**S6 File. The RMSF graphs of EP300 in all ligand and native ligand-bound structures.**
(PDF)

## Acknowledgments

We thank the Tropical Biopharmaca Research Center, IPB University, for supporting this research.

## Author Contributions

**Conceptualization:** Isnatin Miladiyah, Wisnu Ananta Kusuma.

**Data curation:** Rizka Fatriani, Firda Agustin Kartika Pratiwi, Dewi Anggraini Septaningsih, Siska Andrina Kusumastuti, Mochammad Arfin Fardiansyah Nasution, Donny Ramadhan.

**Investigation:** Rizka Fatriani, Firda Agustin Kartika Pratiwi, Siska Andrina Kusumastuti, Mochammad Arfin Fardiansyah Nasution, Donny Ramadhan.

**Methodology:** Annisa Annisa, Dewi Anggraini Septaningsih, Siska Andrina Kusumastuti, Wisnu Ananta Kusuma.

**Supervision:** Annisa Annisa, Sandra Arifin Aziz, Wisnu Ananta Kusuma.

**Visualization:** Rizka Fatriani, Mochammad Arfin Fardiansyah Nasution, Donny Ramadhan.

**Writing – original draft:** Rizka Fatriani, Firda Agustin Kartika Pratiwi.

**Writing – review & editing:** Annisa Annisa, Dewi Anggraini Septaningsih, Sandra Arifin Aziz, Isnatin Miladiyah, Wisnu Ananta Kusuma.

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
