## [Decision Letter · Decision Letter 0]

9 Feb 2024

PONE-D-23-41424Unveiling Kemuning’s (Murraya paniculata) Anti-Obesity Potential: A Network Pharmacology ApproachPLOS ONE

Dear Dr. Kusuma,

Thank you for submitting your manuscript to PLOS ONE. After careful consideration, we feel that it has merit but does not fully meet PLOS ONE’s publication criteria as it currently stands. Therefore, we invite you to submit a revised version of the manuscript that addresses the points raised during the review process.

**ACADEMIC EDITOR: **The authors must address all concerns raised by the reviewers, particularly by providing molecular dynamics simulation data to substantiate the molecular docking studies, and presenting in vitro and in vivo experimental data to support the stated claims. 

We look forward to receiving your revised manuscript.

Kind regards,

Md. Ashraful Hasan, Ph.D.

Academic Editor

PLOS ONE

Journal Requirements:

"The initials author: WAK

Grant numbers: This research was supported by Ministry of Research, Technology and Higher Education, Indonesia, under Competitive Research Grant from Directorate of Higher Education, Indonesia, 2023 No. 102/E5/PG.02.00.PL/2023

URL: https://dikti.kemdikbud.go.id/"

Reviewers' comments:

Reviewer's Responses to Questions

**Comments to the Author**

1. Is the manuscript technically sound, and do the data support the conclusions?

Reviewer #1: Yes

Reviewer #2: Yes

Reviewer #3: No

Reviewer #4: No

2. Has the statistical analysis been performed appropriately and rigorously

Reviewer #1: Yes

Reviewer #2: N/A

Reviewer #3: Yes

Reviewer #4: No

3. Have the authors made all data underlying the findings in their manuscript fully available?

Reviewer #1: Yes

Reviewer #2: Yes

Reviewer #3: Yes

Reviewer #4: Yes

4. Is the manuscript presented in an intelligible fashion and written in standard English?

Reviewer #1: Yes

Reviewer #2: Yes

Reviewer #3: Yes

Reviewer #4: No

5. Review Comments to the Author

Reviewer #1: Murraya paniculata leaves water and ethanol extracts, active compounds were identified and protein targets for obesity, wherein molecular docking where done for PPARG and EP300 protein in the study.

The study has been well structured and results have been elucidated well.

However, the following points has to be considered:-

1) Ten compounds have been identified to have good binding affinity to both the proteins but does all the compounds have anti-obesity properties in other studies.

2) To confirm the molecular docking studies, a molecular dynamics simulation has to done to elucidate it.

Reviewer #2: Although LC-MS results are given in tabular form, the chromatogram is missing, which may be provided.

A molecular dynamics study for understanding the stability of compound-target interaction could add further support.

In-vitro or in-vivo studies to support predicted bioactivity would add value.

Line #42 INSTEAD OF energy entering the body is lower than the energy expended CONSIDER energy entering the body is higher than the energy expended

Line #75 INSTEAD OF drenergic receptor agonists CONSIDER adrenergic receptor agonists

Line #94 INSTEAD OF The anti-obesity effect of M. paniculata as been reported CONSIDER The anti-obesity effect of M. paniculata has been reported

Line # 95 INSTEAD OF 800 mg/kg bb CONSIDER 800 mg/kg

Line #96 INSTEAD OF L-glutamate 2 g/kgbb CONSIDER L-glutamate 2 g/kg

Line # 97 INSTEAD OF An effective dose of 200 mg/kg CONSIDER An effective dose of 200 mg/kg

Line # 366 INSTEAD OF ten active compounds CONSIDER Ten potentially active compounds

Line # 370 INSTEAD OF Various methods have been used for obesity treatment, including drug discovery from herbal….. CONSIDER Various methods have been used for obesity treatment, including herbal..

Line # 468 INSTEAD OF possible mechanism of the active compounds of M. paniculata CONSIDER possible mechanism of the compounds of M. paniculata

Line # 469 INSTEAD OF in curing obesity CONSIDER in treatment/management of obesity.

Reviewer #3: The authors seem to present a in silico analysis of the Kemuning’s (Murraya paniculata) Anti-Obesity Potential. however I would recommed major revision of the manuscript in its present form :-

1.Lack of Novelty:

- The integration method described in the paper does not seem to introduce any novel approach or technique. Similar methodologies have been employed in previous studies, and the paper fails to highlight any substantial advancement in the field.

2. Ambiguity in Results Interpretation:

- The paper lacks clarity in interpreting the results obtained from network pharmacology and molecular docking. The significance of the identified targets (EP300, PPARG, and PPARGC1A) is not well-established, and the connections between these targets and the treatment of obesity are inadequately explained.

3. Incomplete Validation:

- The paper lacks experimental validation of the proposed targeting mechanism. While molecular docking results are presented, there is a lack of in vitro or in vivo experiments to support the claims made regarding the binding affinity and the effectiveness of the active compounds from M. paniculata.

4. Limited Scope of Study with Over generalizations

- The scope of the study appears to be limited to the identification of potential targets and molecular interactions. However, a comprehensive understanding of the biological mechanisms involved in treating obesity requires additional investigations, such as cellular assays and animal model studies, which are lacking in this paper. he conclusions drawn from the study, particularly the statement that M. paniculata treats obesity by targeting PPARG and EP300 proteins, seem overly generalized. The results should be cautiously interpreted and discussed within the context of the study's limitations and uncertainties.

7. Inadequate Comparative Analysis:

- The paper lacks a comprehensive comparison with existing literature on the topic. A thorough review and discussion of similar studies would provide a clearer understanding of how the current work contributes to the existing body of knowledge. Some statements lack proper citation to support the claims made. It is essential to provide a robust foundation for the research by referencing relevant studies and findings in the field.

8. Language and Writing Style:

- The manuscript suffers from issues related to language and writing style. The narrative is convoluted at times, making it challenging for readers to follow the logical flow of the research.

Recommendation for Major Revisions:

- Given the aforementioned concerns, it is recommended that the paper undergoes major revisions to address the identified issues and strengthen the overall quality and validity of the study.

Reviewer #4: This manuscript is very unorganized and unplanned, and it is very hard to understand what authors actually want to show from their study. There is no alignment between the study flow they showed in Figure 3 and what they described in the results section.

First of all, network pharmacology should be made based on the M. paniculata metabolites identified from the LC-MS, and then the relevant targets and signaling pathways must be validated by experimental studies. but authors made their network based on the metabolites found in IJAH Analytics database, which makes their network prediction technically and scientifically not sound. Writing in both the result and discussion sections is also very crude, and docking is not enough to prove interaction. Furthermore, the authors did not prove any relationship between the M. paniculata network and obesity. The roles of their identified targets in obesity pathology and whether those compounds can ameliorate obesity must be proved experimentally.

6. PLOS authors have the option to publish the peer review history of their article (what does this mean?). If published, this will include your full peer review and any attached files.

Reviewer #1: **Yes: **Dicky John Davis G

Reviewer #2: No

Reviewer #3: **Yes: **Subhadip Banerjee

Reviewer #4: No

---

## [Author Response · Author response to Decision Letter 0]

8 Apr 2024

Original Manuscript ID: PONE-D-23-41424

Original Article Title: Unveiling Kemuning’s (Murraya paniculata) Anti-Obesity Potential: A Network Pharmacology Approach

To: PLOS ONE Editor

Re: Response to Reviewers

Dear Editor,

Thank you for allowing us to resubmit our manuscript and giving us the opportunity to respond to the reviewers’ comments. We appreciate the time and effort that you and the reviewers have dedicated to providing valuable comments.

After careful revision, we hope that the manuscript meets your publication criteria.

Below, we provide our point-by-point responses to the comments (Response to Reviewers), an updated manuscript with yellow highlights (Revised Manuscript with Track Changes), and an updated manuscript without tracks (Manuscript).

Thank you. Best regards,

Wisnu Ananta Kusuma, et al.

 

Editor

Author Response: Thank you for pointing this out. We are currently following the PLOS ONE template.

Author Response: Thank you for your suggestion. We have added the grant numbers to the funding information. “This research was supported by the Ministry of Research, Technology and Higher Education, Indonesia, under the Competitive Research Grant from the Directorate of Higher Education, Indonesia, in 2023. No. 102/E5/PG.02.00.PL/2023. URL: https://dikti.kemdikbud.go.id/. The initial of the author who received the grant is WAK. The funders had no role in the study design, data collection and analysis, decision to publish, or manuscript preparation.” This information is provided in line 665. 

"The initials author: WAK Grant numbers: This research was supported by Ministry of Research, Technology and Higher Education, Indonesia, under Competitive Research Grant from Directorate of Higher Education, Indonesia, 2023 No. 102/E5/PG.02.00.PL/2023 URL: https://dikti.kemdikbud.go.id/"

Author Response: We have equalized the grant information in 'Funding Information' and 'Financial Disclosure' and added the role of funders in this study. "The funder (the Ministry of Research, Technology and Higher Education, Indonesia) had no role in the research design, data collection and analysis, decision to publish, or preparation of the manuscript." This information can be found in line 669.

Reviewer #1

Murraya paniculata leaves water and ethanol extracts, active compounds were identified and protein targets for obesity, wherein molecular docking where done for PPARG and EP300 protein in the study.

The study has been well structured and results have been elucidated well.

However, the following points has to be considered:-

1. Ten compounds have been identified to have good binding affinity to both the proteins but does all the compounds have anti-obesity properties in other studies.

Author Response: We have added other studies related to the compound obtained from metabolite profiling to strengthen the statement regarding its potential as an anti-obesity agent. The elucidation is in lines 595 to 617.

2. To confirm the molecular docking studies, a molecular dynamics simulation has to done to elucidate it.

Author Response: Thank you for your feedback regarding the elucidation of our molecular docking results. In response to your concern, we have carried out molecular dynamics simulations to elucidate the molecular docking results. The molecular dynamics results can be seen in Figs 8 and 9; the results are on lines 365 to 413. Adding molecular dynamics data strengthens the molecular docking data for the stability of protein-ligand binding.

Reviewer #2

1. Although LC-MS results are given in tabular form, the chromatogram is missing, which may be provided.

Author Response: Thank you for your suggestion. We have added a chromatogram graph in Fig 7—the base peak chromatogram in positive ionization mode for the ethanol extract of M. paniculata. The chromatogram is shown in line 333.

2. A molecular dynamics study for understanding the stability of compound-target interaction could add further support.

Author Response: Thank you for such an insightful suggestion about the molecular dynamic simulation. We have already added a molecular dynamics simulation to improve our research. The molecular dynamic results can be seen in Figs 8 and 9; the results are on lines 365 to 413. 

3. In-vitro or in-vivo studies to support predicted bioactivity would add value.

Author Response: We appreciate your suggestion. To improve the evidence for the potential of M. paniculata as an anti-obesity agent. We performed in vitro tests using 3T3-L1 cells. We have presented the data from this in vitro research in the revised manuscript. The results are shown in lines 414–448.

4. Line #42 INSTEAD OF energy entering the body is lower than the energy expended CONSIDER energy entering the body is higher than the energy expended

Author Response: We agree with your suggestion, and we changed it to “the energy consumed is higher than that expended.” This information is provided in line 55.

5. Line #75 INSTEAD OF drenergic receptor agonists CONSIDER adrenergic receptor agonists

Author Response: Thank you for your suggestion. We have already updated the Introduction section of the manuscript and deleted this sentence.

6. Line #94 INSTEAD OF The anti-obesity effect of M. paniculata as been reported CONSIDER The anti-obesity effect of M. paniculata has been reported

Author Response: Thank you for your correction; we have already changed it to “The anti-obesity effect of M. paniculata has been reported.” This information is provided in line 78.

7. Line # 95 INSTEAD OF 800 mg/kg bb CONSIDER 800 mg/kg

Author Response: We have revised this. This information is provided in line 79.

8. Line #96 INSTEAD OF L-glutamate 2 g/kgbb CONSIDER L-glutamate 2 g/kg

Author Response: We agree with your comment; the revised manuscript is provided in line 80.

9. Line # 97 INSTEAD OF An effective dose of 200 mg/kg CONSIDER An effective dose of 200 mg/kg

Author Response: Thank you for your suggestion. The revision is provided in line 81.

10. Line # 366 INSTEAD OF ten active compounds CONSIDER Ten potentially active compounds

Author Response: We have already changed to “potentially active compounds.” This information is provided in line 464.

11. Line # 370 INSTEAD OF Various methods have been used for obesity treatment, including drug discovery from herbal….. CONSIDER Various methods have been used for obesity treatment, including herbal..

Author Response: Thank you for your comment. We have already updated the Discussion section of the manuscript and deleted this sentence.

12. Line # 468 INSTEAD OF possible mechanism of the active compounds of M. paniculata CONSIDER possible mechanism of the compounds of M. paniculata

Author Response: Thank you for your comment. We have updated the manuscript conclusion and deleted this sentence. 

13. Line # 469 INSTEAD OF in curing obesity CONSIDER in treatment/management of obesity.

Author Response: Regarding your comment, we have revised it to treatment and management. The revised manuscript is in line 619.

Reviewer #3

The authors seem to present a in silico analysis of the Kemuning’s (Murraya paniculata) Anti-Obesity Potential. however I would recommed major revision of the manuscript in its present form :-

1. Lack of Novelty:

- The integration method described in the paper does not seem to introduce any novel approach or technique. Similar methodologies have been employed in previous studies, and the paper fails to highlight any substantial advancement in the field.

Author Response: Thank you for your valuable suggestion. The novelty of our study is that we used the skyline query technique, an optimization technique developed from the maxima point problem, and implemented the block nested loop (BNL) algorithm to obtain dominant protein targets from protein-protein interaction networks. 

2. Ambiguity in Results Interpretation:

- The paper lacks clarity in interpreting the results obtained from network pharmacology and molecular docking. The significance of the identified targets (EP300, PPARG, and PPARGC1A) is not well-established, and the connections between these targets and the treatment of obesity are inadequately explained.

Author Response: Thank you for your insightful comment. In the first version, the explanation of the connections between these targets and the treatment of obesity is inadequate. Therefore, we have updated this revised manuscript. We have added molecular dynamic simulations and in vitro testing to strengthen the statement regarding the potential of M. paniculata to treat obesity. We discovered that M. paniculata extract can decrease PPARG and EP300 gene expression. 

3. Incomplete Validation:

- The paper lacks experimental validation of the proposed targeting mechanism. While molecular docking results are presented, there is a lack of in vitro or in vivo experiments to support the claims made regarding the binding affinity and the effectiveness of the active compounds from M. paniculata.

Author Response: We really appreciate your suggestion. We performed in vitro tests using 3T3-L1 cells to experimentally validate the molecular docking results. We have presented the data from this in vitro test in the revised manuscript. The results are on lines 414–448.

4. Limited Scope of Study with Over generalizations

- The scope of the study appears to be limited to the identification of potential targets and molecular interactions. However, a comprehensive understanding of the biological mechanisms involved in treating obesity requires additional investigations, such as cellular assays and animal model studies, which are lacking in this paper. he conclusions drawn from the study, particularly the statement that M. paniculata treats obesity by targeting PPARG and EP300 proteins, seem overly generalized. The results should be cautiously interpreted and discussed within the context of the study's limitations and uncertainties.

Author Response: We appreciate your comment. To improve the understanding and purpose of this research, we have already conducted an in vitro assay to confirm the potency of M. paniculata as an anti-obesity agent. We treated preadipocyte 3T3-L1 cells with various concentrations of M. paniculata extract. Adipogenesis-inducing genes in mature 3T3-L1 adipocytes were quantified to determine the obesity potential of the extract.

5. Inadequate Comparative Analysis:

- The paper lacks a comprehensive comparison with existing literature on the topic. A thorough review and discussion of similar studies would provide a clearer understanding of how the current work contributes to the existing body of knowledge. Some statements lack proper citation to support the claims made. It is essential to provide a robust foundation for the research by referencing relevant studies and findings in the field.

Author Response: To address this concern, we have added other research studies to provide an understanding of the results of this research. We have also updated our manuscript and added molecular dynamics simulations and in vitro tests to this manuscript. Molecular dynamics simulations and in vitro tests can elucidate network pharmacology and molecular docking results.

6. Language and Writing Style:

- The manuscript suffers from issues related to language and writing style. The narrative is convoluted at times, making it challenging for readers to follow the logical flow of the research.

Recommendation for Major Revisions:

- Given the aforementioned concerns, it is recommended that the paper undergoes major revisions to address the identified issues and strengthen the overall quality and validity of the study.

Author Response: We appreciate your feedback. We have updated the manuscript and conducted proofreading in English to improve the quality of the manuscript in terms of language. Following is the proofread certificate.

Reviewer #4

This manuscript is very unorganized and unplanned, and it is very hard to understand what authors actually want to show from their study. There is no alignment between the study flow they showed in Figure 3 and what they described in the results section.

First of all, network pharmacology should be made based on the M. paniculata metabolites identified from the LC-MS, and then the relevant targets and signaling pathways must be validated by experimental studies. but authors made their network based on the metabolites found in IJAH Analytics database, which makes their network prediction technically and scientifically not sound. Writing in both the result and discussion sections is also very crude, and docking is not enough to prove interaction. Furthermore, the authors did not prove any relationship between the M. paniculata network and obesity. The roles of their identified targets in obesity pathology and whether those compounds can ameliorate obesity must be proved experimentally.

Author Response: Thank you for pointing this out. The aim of this study was to demonstrate the pharmacological mechanism of M. paniculata as a potential anti-obesity agent. As mentioned by the reviewer, the previous studies started by searching for target proteins and predicting the interaction of active compounds in M. paniculata using SwissTargetPrediction (http://swisstargetprediction.ch/). However, we would like to show another different approach or perspective from previous studies. Our approach used the principle of big data analysis. Therefore, we started by selecting protein targets related to obesity by collecting data from various sources, including the OMIM database (https://www.omim.org/) and IJAH Analytics (https://www.ijah.apps.cs.ipb.ac.id), and then analyzed them to obtain the target proteins that have the most important role in obesity. IJAH is a web application and database that shows the plant-compound-protein-disease relationship. Plant data from IJAH were obtained from plants that grow in Indonesia, including M. paniculata, and were also obtained from KNApSack (http://www.knapsackfamily.com/KNApSAcK/). The interactions of compounds and proteins in IJAH were obtained from PubChem and Uniprot; the data means that it has been validated/curated, so it is not based on the predicted results. With this approach, we hope to obtain more accurate target protein screening results that represent obesity by applying a skyline query. Therefore, the compounds in M. paniculata LCMS results that only interact with these selected target proteins were validated and further analyzed through molecular docking dynamics, molecular simulation, and in vitro. We updated this manuscript. To understand the purpose of this research, we added a dynamics molecular simulation and an in vitro assay to prove the relationship between M. paniculata, targets from network pharmacology, and the potential as anti-obesity.

---

## [Decision Letter · Decision Letter 1]

3 Jun 2024

Unveiling the Anti-Obesity Potential of Kemuning (Murraya paniculata): A Network Pharmacology Approach

PONE-D-23-41424R1

Dear Dr. Kusuma,

We’re pleased to inform you that your manuscript has been judged scientifically suitable for publication and will be formally accepted for publication once it meets all outstanding technical requirements. Please note that the comments from Reviewer 2 are irrelevant to this article and do not need to be addressed by the authors. Additionally, the manuscript does not mention the source of the plant leaves used in the experiments (line 158 onwards). Therefore, the authors are advised to include the source of the plant leaves to aid in the reproducibility of the methodology.

Kind regards,

Md. Ashraful Hasan, Ph.D.

Academic Editor

PLOS ONE

Reviewers' comments:

Reviewer's Responses to Questions

**Comments to the Author**

1. If the authors have adequately addressed your comments raised in a previous round of review and you feel that this manuscript is now acceptable for publication, you may indicate that here to bypass the “Comments to the Author” section, enter your conflict of interest statement in the “Confidential to Editor” section, and submit your "Accept" recommendation.

Reviewer #1: All comments have been addressed

Reviewer #2: (No Response)

Reviewer #3: All comments have been addressed

2. Is the manuscript technically sound, and do the data support the conclusions?

Reviewer #1: Yes

Reviewer #2: Yes

Reviewer #3: Yes

3. Has the statistical analysis been performed appropriately and rigorously? 

Reviewer #1: Yes

Reviewer #2: Yes

Reviewer #3: Yes

4. Have the authors made all data underlying the findings in their manuscript fully available?

Reviewer #1: Yes

Reviewer #2: Yes

Reviewer #3: Yes

5. Is the manuscript presented in an intelligible fashion and written in standard English?

Reviewer #1: Yes

Reviewer #2: Yes

Reviewer #3: Yes

6. Review Comments to the Author

Reviewer #1: Murraya paniculata leaves water and ethanol extracts, active compounds were identified and protein targets for obesity, wherein molecular docking and dynamics simulation where done for PPARG and EP300 protein in the study and complimented with in-vitro studies.

Authors have adequately addressed the comments raised in a previous round of review such as

1) LC-MS chromatogram has been provided

2) Molecular dynamics study was also done.

3) In-vitro studies using 3T3-L1 cells have been conducted.

Revised manuscript has shown that M.paniculata extract can decrease PPARG and EP300 gene expression.

Reviewer #2: The manuscript attempts to identify the risk factors associated with the nutritional status of 0-59 years children in Ethiopia.

The manuscript presents data required for the analysis and have cited the source. I have reviewed the manuscript with the help of colleagues from Community Medicine and Biostatistics and have the following suggestions to make.

1. The key-words may be arranged alphabetically.

2. In the background, the burden of undernutrition documented by previous studies needs to be added.

Any nutritional programme, if ongoing or carried out in the past should also be detailed for better understanding.

3. In Methods, a brief description of study area and settings is desired since this information is necessary for better understanding by readers outside of the region and it is often cumbersome to go through the reports.

4. Since the study design is cross-sectional, it is preferable to use the prevalence ratio (PP) as the effect estimate rather than the odds ratio. How can the authors justify this choice? (Line 182).

5. Results. The manuscript displays inconsistency in the number of digits after decimal points. Some sections feature only one digit, while others have two. Authors may like to use uniform decimal places across all sections.

6. There are some errors, perhaps typographical in the manuscript e.g Page 15, Line 365, 'to'?

7. References: The manuscript should be checked for uniformity of citation. Some references does not show month and some citation dates missing (eg Ref 15, 16, 19,20).

Reviewer #3: (No Response)

7. PLOS authors have the option to publish the peer review history of their article (what does this mean?). If published, this will include your full peer review and any attached files.

Reviewer #1: **Yes: **Dicky John Davis G

Reviewer #2: No

Reviewer #3: **Yes: **Subhadip banerjee

---

## [Editor Report · Acceptance letter]

7 Jul 2024

PONE-D-23-41424R1 

PLOS ONE

Dear Dr. Kusuma, 

I'm pleased to inform you that your manuscript has been deemed suitable for publication in PLOS ONE. Congratulations! Your manuscript is now being handed over to our production team.

Kind regards, 

on behalf of

Dr. Md. Ashraful Hasan 

Academic Editor

PLOS ONE